# Single-molecule analysis of the entire perfringolysin O pore formation pathway

Conall McGuinness[1†], James C Walsh[1*†], Charles Bayly-Jones[2], Michelle A Dunstone[2], Michelle P Christie[3], Craig J Morton[3‡], Michael W Parker[3,4], Till Böcking[1*]

[1]EMBL Australia Node in Single Molecule Science, School of Biomedical Sciences, University of New South Wales, Sydney, Australia; [2]Biomedicine Discovery Institute, Department of Biochemistry and Molecular Biology, Monash University, Melbourne, Australia; [3]Department of Biochemistry and Pharmacology, Bio21 Molecular Science and Biotechnology Institute, University of Melbourne, Victoria, Australia; [4]Structural Biology Unit, St. Vincent's Institute of Medical Research, Victoria, Australia

**\*For correspondence:**
james.walsh@unsw.edu.au
(JCW);
till.boecking@unsw.edu.au (TB)

[†]These authors contributed equally to this work

**Present address:** [‡]CSIRO Biomedical Manufacturing Program, Victoria, Australia

**Competing interest:** The authors declare that no competing interests exist.

**Abstract** The cholesterol-dependent cytolysin perfringolysin O (PFO) is secreted by *Clostridium perfringens* as a bacterial virulence factor able to form giant ring-shaped pores that perforate and ultimately lyse mammalian cell membranes. To resolve the kinetics of all steps in the assembly pathway, we have used single-molecule fluorescence imaging to follow the dynamics of PFO on dye-loaded liposomes that lead to opening of a pore and release of the encapsulated dye. Formation of a long-lived membrane-bound PFO dimer nucleates the growth of an irreversible oligomer. The growing oligomer can insert into the membrane and open a pore at stoichiometries ranging from tetramers to full rings (~35 mers), whereby the rate of insertion increases linearly with the number of subunits. Oligomers that insert before the ring is complete continue to grow by monomer addition post insertion. Overall, our observations suggest that PFO membrane insertion is kinetically controlled.

## Editor's evaluation

This paper presents a detailed single-molecule, multi-color microscopy study of the real-time assembly of perfringolysin O, a member of the membrane attack complex perforin cholesterol-dependent cytolysin superfamily. With the ability to resolve different reaction species simultaneously with membrane leakage, this work provides key mechanistic details including identifying assemblies involved in membrane lysis, and how membrane binding, oligomerization, and pore transitioning depends on concentration and pH. This study will be of interest to many, particularly those studying cytolysin mechanisms, but also the broader field of single-molecule studies of membrane binding proteins.

## Introduction

Pore-forming proteins (PFPs) possess an ancient and ubiquitous mechanism for forming aqueous channels in the membranes surrounding cells and organelles (*Dal Peraro and van der Goot, 2016*; *Johnstone et al., 2021*). The largest and most sequence diverse class of PFPs is the membrane attack complex-perforin (MACPF)/cholesterol dependent cytolysin (CDC) superfamily with thousands of members now identified (*Christie et al., 2018*; *Dunstone and Tweten, 2012*; *Rosado et al., 2008*). Members of the MACPF family are found in all kingdoms of life but are most well characterised as effectors in the vertebrate immune system. Conversely, CDCs are bacterial virulence and defence

factors (*Tweten, 2005*). While these two families differ greatly in sequence, they are linked together by a highly conserved 3D fold, which drives oligomerisation into rings of 12–40 PFP monomers and subsequent membrane insertion, ultimately forming a membrane-spanning β-barrel (*Shatursky et al., 1999*). The open pore has an unusually large lumen of ~25–30 nm in the case of CDCs, allowing the passive transport of folded proteins across membranes. (*Tweten et al., 2001*).

Perfringolysin O (PFO), a prototypical example of a CDC, is secreted by the anaerobic bacterium *Clostridium perfringens* (*Tweten et al., 2001*), which is involved in the development of gas gangrene and necrohemorrhagic enteritis (*Awad et al., 2001*; *Verherstraeten et al., 2015*; *Verherstraeten et al., 2013*). As with the majority of CDCs, PFO binds to cholesterol-rich membranes and oligomerises to form large (25–30 nm) doughnut-shaped pores, ultimately leading to cell lysis (*Dang et al., 2005*; *Tilley et al., 2005*). A small subset of CDCs do not require cholesterol for binding (*Giddings et al., 2004*; *Ragaliauskas et al., 2019*); however, it remains necessary for membrane insertion (*Jacobs et al., 1998*; *Polekhina et al., 2005*).

The steps in canonical CDC pore-formation have been well characterised; the proteins are secreted as soluble monomers, monomers bind to and then oligomerise on target membranes to form a ring-shaped prepore complex, the prepore complex then undergoes a concerted conformational change and inserts in the membrane to form a large (25–30 nm) amphipathic β-barrel bilayer-spanning pore (*Figure 1A*; *Morton et al., 2019*). To resolve these steps, point mutations of PFO have been used to investigate kinetically trapped intermediates alongside fluorescent conjugates acting as environmental indicators (*Evans and Tweten, 2021*; *Ramachandran et al., 2004*), with some intermediates observed by atomic force microscopy (AFM) (*Czajkowsky et al., 2004*) and electron microscopy (EM) (*van Pee et al., 2017*) high-resolution imaging. These and other studies have defined the key molecular events and rearrangements required for pore formation. Initially, membrane binding is mediated by domain 4 (D4) which specifically recognises cholesterol in the lipid bilayer. Interdomain contacts in the CDC fold drive oligomerisation forming arc prepores. CDCs undergo a drastic collapse by rotation of domain 2 (D2) which lowers domain 3 (D3) toward the membrane, enabling the α-helical bundles (αHB) to unfurl and insert into the bilayer forming transmembrane β-hairpins. Notably, these data showed the prevalence of full prepore and inserted rings and assumed them to be the functional mechanism of pore formation, although incomplete arc-shaped oligomers have been observed by other labs (*Leung et al., 2014*; *Sonnen et al., 2014*).

Liposome dye release assays have long been used to investigate the kinetics of PFO pore formation (*Evans et al., 2020*; *Heuck et al., 2000*; *Shepard et al., 2000*). This assay measures the rate of release of a fluorescent dye from liposomes in bulk after incubation with PFO, which is correlated to the rate of pore formation. The main shortcoming of these experiments is that they give a single bulk readout of the reaction over time. The intrinsically stochastic nature of nucleation makes it impossible to synchronise pore formation on multiple liposomes even if they are exposed to PFO at the same time. As a result, any ensemble measurement of PFO binding is blurred by averaging the growth of pores at different stages of formation. Similarly, pore insertion is the culmination of multiple stochastic processes, occurring asynchronously between liposomes. As such, it is difficult to identify the specific molecular interaction in which variation underpins any observed changes in the bulk measurement.

To overcome the limitation of ensemble averaging, imaging methods have been developed to follow PFP assembly at the level of individual pores (*Benke et al., 2015*; *Ros et al., 2021*; *Sathyanarayana et al., 2018*). Imaging modalities applied to CDC assembly on planar lipid bilayers include high speed atomic force microscopy, as shown for suilysin (*Leung et al., 2014*) and listeriolysin O (*Ruan et al., 2016*) and single-molecule fluorescence tracking, as shown for PFO assembly on a droplet interface bilayer (*Senior et al., 2022*). These imaging studies support the insertion of incomplete arc-shaped membrane lesions, suggesting an alternative mechanism of pore formation that is distinct from the canonical prepore formation prior to insertion. This observation raises the question as to when and how release of the membrane spanning regions is triggered, which cannot be correlated with key assembly steps using ensemble methods. Related to this matter, it is also unclear whether release and insertion of the membrane spanning regions from each of the subunits occurs in a concerted or sequential fashion.

To observe PFP assembly and membrane permeabilisation simultaneously at the single molecule level, we have adapted the liposome dye release assay to measure kinetics on individual liposomes. By using single molecule total internal reflection fluorescence (TIRF) microscopy, we visualised the

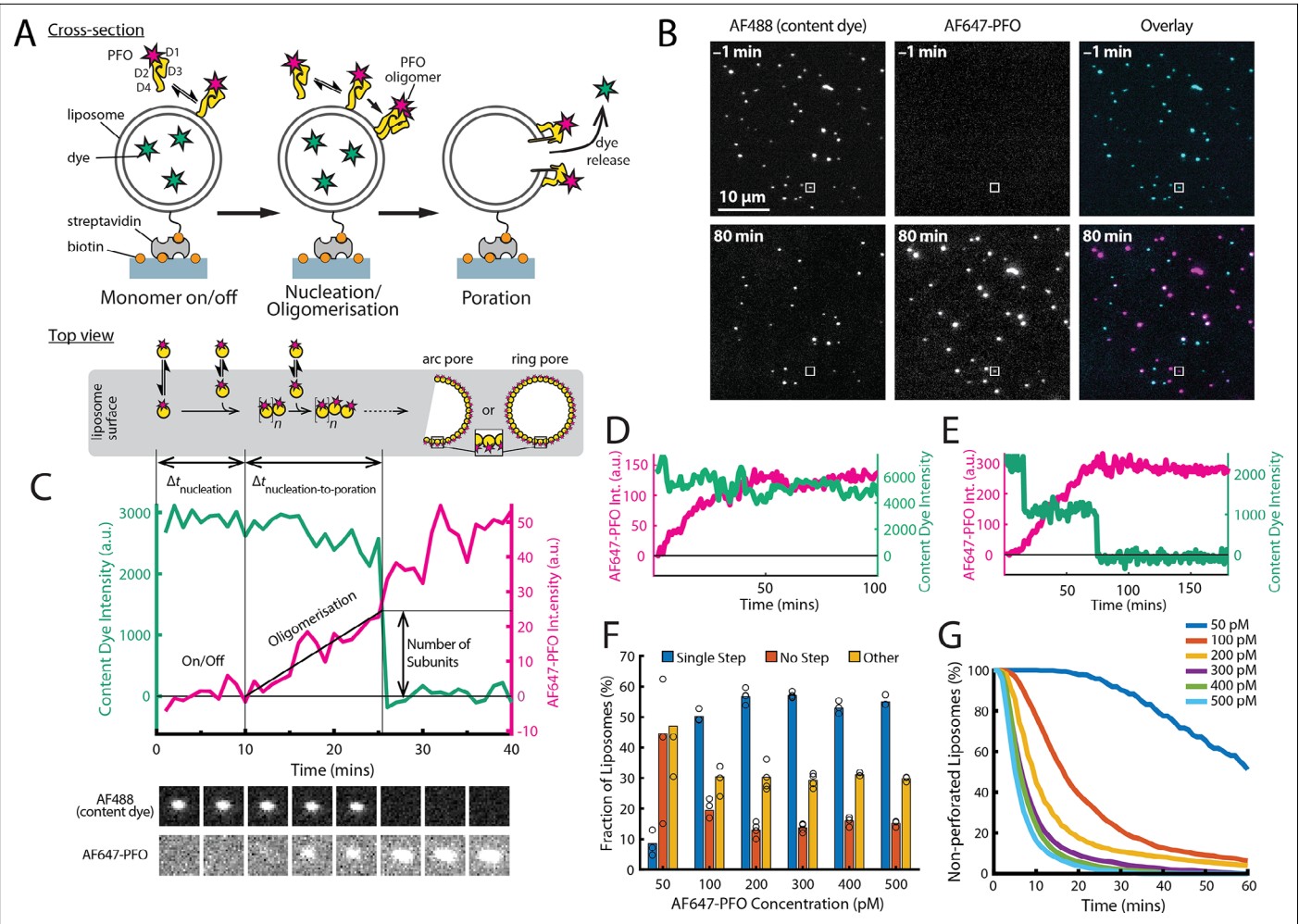

**Figure 1.** PFO pore formation assay. (**A**) Schematic of the TIRF assay to measure the PFO assembly and membrane pore formation. Liposomes loaded with AF488 as a content marker are bound to a coated glass coverslip at the bottom of a microfluidic device. AF647-labelled PFO injected into the flow channel reversibly binds to the liposome, assembles into an oligomer and ultimately forms an open arc or ring pore releasing the encapsulated dye. (**B**) TIRF images (400×400 pixel region) from a pore formation time series before (–1 min) and after (80 min) addition of 100 pM AF647-PFO to immobilised liposomes showing the AF488 channel (left), the AF647-PFO channel (middle) and an overlay of both channels (right; AF488 in cyan and AF647-PFO in magenta). (**C**) Example fluorescence intensity traces recorded at the location of the single liposome (marked with the white box in panel B) in the content dye (blue-green) and AF647-PFO (magenta) channels. Dye release pinpoints the time of membrane permeabilisation. Snapshots (1.7 µm×1.7 µm) of the corresponding liposome in both channels are shown below the traces. Additional example traces are shown in *Figure 1—figure supplement 3*. (**D**) Example "No Step" trace where the content dye intensity (green) remains constant throughout the experiment (**E**) Example 'Other' trace where the dye content dye intensity (green) decreases in multiple steps. (**F**) Fraction of liposomes with content dye traces classified on the basis of step fitting as (1) single step, (2) no dye loss (<25% decrease in intensity) or (3) other release profiles (includes traces with multiple steps or incomplete dye loss). Example traces for each of these classes are shown in *Figure 1—figure supplement 3*. (**G**) Distributions of single-step dye release times from liposomes at AF647-PFO concentrations between 50–500 pM. Number of replicates in (**F**) and (**G**): 3 experiments for 50 pM, 100 pM, 400 pM, 500 pM; 4 experiments for 200 pM, 300 pM. All experiments were conducted at room temperature using a 20 mM HEPES buffer (pH 7) containing 100 mM NaCl and 0.01 mg/mL BSA.

The online version of this article includes the following figure supplement(s) for figure 1:

**Figure supplement 1.** Liposome size distribution.

**Figure supplement 2.** Fluorescence labelling of PFO does not impair pore formation activity.

**Figure supplement 3.** Classification of single-liposome traces on the basis of the content dye release profile.

**Figure supplement 4.** Theoretical computation of the dye release rate following pore opening.

binding, nucleation, build up and insertion of individual PFO species. These data were subsequently used to develop a mathematical model for PFO pore formation in which parameters can predict the number of subunits in PFO oligomers at the time of insertion. We have also found in our assay that inserted PFO arcs, currently thought to be kinetically trapped, can continue to grow post insertion.

## Results

### Fluorescence imaging of PFO pore assembly kinetics on single liposomes

Here, we developed a single-molecule approach to observe in real time the dynamic interactions between PFO and dye-loaded liposomes leading to dye release from individual liposomes using total internal reflection fluorescence (TIRF) microscopy. The assay design is shown schematically in *Figure 1A*. Large unilamellar liposomes made of a synthetic lipid mixture containing cholesterol, 1-palmitoyl-2-oleoyl-phosphatidylcholine and a small amount of a phosphatidylethanolamine derivative with biotinylated headgroup (55:44:1 molar ratio) were loaded with the small fluorescent dye Alexa Fluor 488 (AF488) during extrusion. The high-cholesterol content facilitates PFO activity on model membranes and is consistent with previous biochemical studies (*Shepard et al., 2000*). The liposomes had an average diameter of ~200 nm (183±37 nm measured by dynamic light scattering, 172±18 nm measured using super resolution microscopy; *Figure 1—figure supplement 1*) and were captured on the surface of a streptavidin-coated glass coverslip at the bottom of a microfluidic channel device. Recombinant cysteine-less PFO (see Appendix) was labelled via lysine residues (*Harris et al., 1991*) with Alexa Fluor 647 (AF647) using NHS ester chemistry, whereby most molecules contained a single dye (*Figure 2E*) and AF647-PFO retained full pore-formation activity (*Figure 1—figure supplement 2*). AF647-PFO was constantly flowed at a defined concentration through the microfluidic channel while imaging AF647-PFO assembly and AF488 release from liposomes by dual-colour time-lapse TIRF microscopy. The immobilised liposomes appeared as diffraction-limited spots in the AF488 channel (*Figure 1B*, top left), which disappeared over the course of the experiment (*Figure 1B*, bottom left). In contrast, AF647-PFO was initially undetectable (*Figure 1B*, top middle), before gradually accumulating in spots that colocalised with the liposomes (*Figure 1B*, bottom middle). Liposomes subjected to the same buffer flow, but in the presence of AF647-PFO concentrations (≤20 pM) below the threshold required for PFO assembly, retained the content dye for at least 7 hr, confirming that dye release was dependent on pore formation (*Figure 2—figure supplement 2*).

TIRF pore formation movies were analysed using automated software for tracking the fluorescence intensity over time at each liposome location in both channels to generate single-liposome AF647-PFO binding and content dye release traces. A typical dual-colour pore formation trace recorded at a single liposome (highlighted with a white box in *Figure 1B*) is shown in *Figure 1C*. A montage of corresponding images of the liposome and AF647-PFO channels are shown below the plot. Initially, the liposome intensity remains high while there is no signal above noise in the AF647-PFO channel. During this phase, PFO monomers interact transiently with the liposome membrane, which can be resolved by imaging at high temporal resolution (as described below) but are not detected in the pore formation traces. Eventually, the AF647-PFO signal rises above background, which we attribute to nucleation of a PFO oligomer that is stably bound to the liposome membrane and continues to grow in number of subunits (and hence intensity). A sudden drop in the content dye signal to background levels pinpoints the time of membrane poration, allowing rapid diffusion of AF488 out of the liposome. We attribute this event to the opening of a single transmembrane pore as a result of a PFO oligomer inserting into the membrane. We also determined the number of labelled subunits in the PFO oligomer forming the open pore structure from the AF647 intensity at the time of dye release divided by the intensity of a single AF647-PFO molecule. Interestingly, the AF647-PFO signal continues to increase beyond the time of pore formation on most liposomes. When experiments were continued long after poration, liposomes continued to bind PFO to levels much greater than the value expected for a single ring-shaped pore (~35 subunits), suggesting that eventually multiple pores form on a single liposome. Further example traces are shown in *Figure 1—figure supplement 3*.

Typically, a field of view contained ~2,300 liposomes (corresponding to a surface density of 0.074 liposomes/µm²). At AF647-PFO concentrations between 100 and 500 pM, most liposomes lost their dye signal in a single step (>50%) (*Figure 1C and F*), while ~30% of dye release traces showed

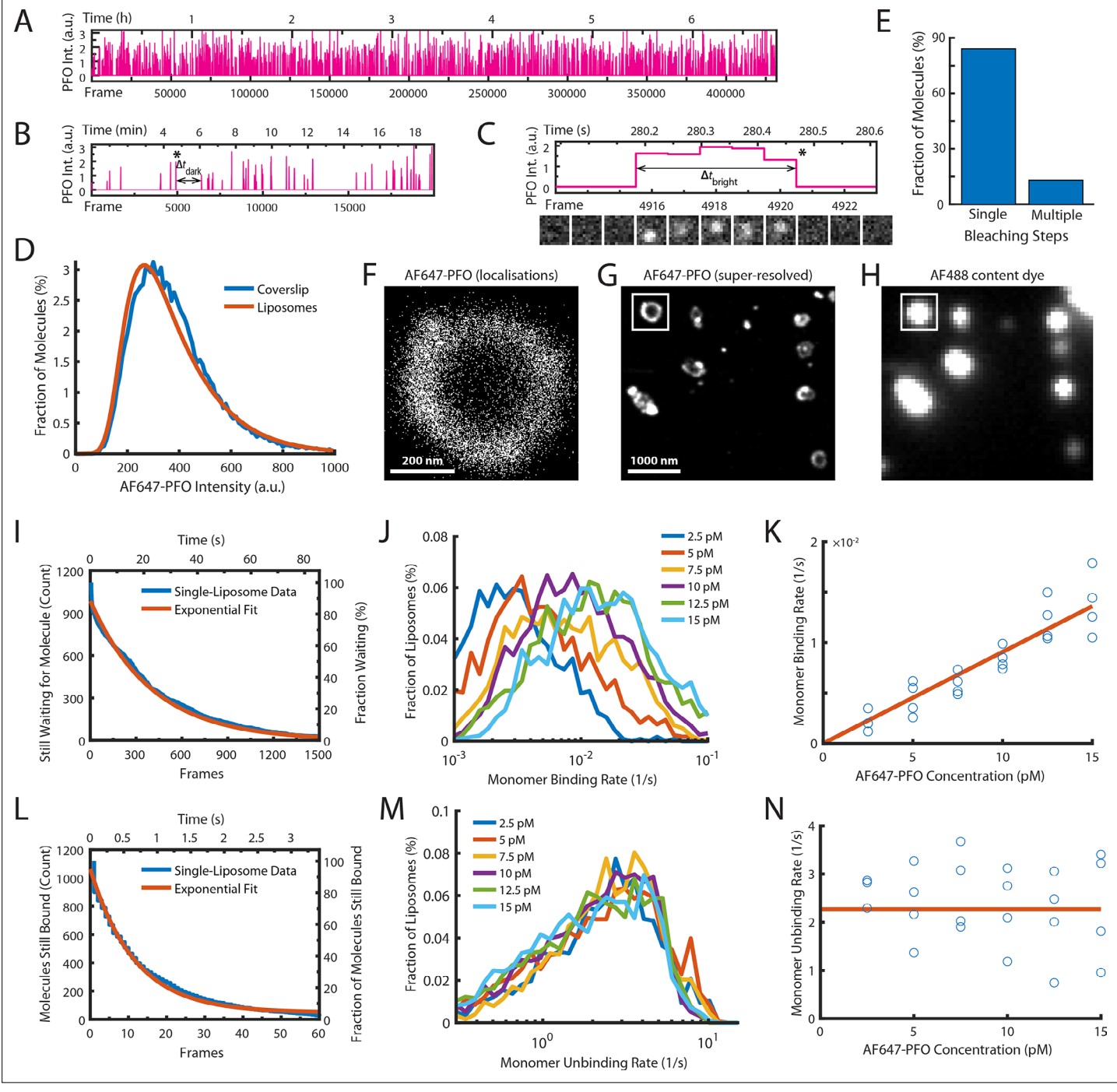

**Figure 2.** Single-molecule PFO binding to liposomes is short-lived. (**A–C**) Trace of single AF647-PFO molecules detected at the location of a single liposome from a time series (432000 frames at 17.5 frames per second) acquired in the presence of 10 pM AF647-PFO. (**A**) Entire trace (6.8 hr). (**B**) Expanded view of the first 20 min. The double arrow indicates the waiting time ($\Delta t_{dark}$) between the peak marked with an asterisk and the subsequent peak. (**C**) Expanded view of the peak marked with an asterisk in B. The double arrow indicates the duration ($\Delta t_{bright}$) of a AF647-PFO molecule on the liposome. Snapshots (1.2 μm × 1.2 μm) of the molecule detected in 5 successive frames are shown below the trace. (**D**) Intensity distribution of AF647-PFO molecules bound to liposomes (orange line) or immobilised directly on a glass coverslip (blue). Intensities were determined from single frames using Picasso. (**E**) Photobleaching analysis of AF647-PFO molecules immobilised on a glass coverslip. The majority (84%) of AF647-PFO bleached in a single step confirming that they are monomeric. (**D**) and (**E**) show the pooled data from 40 fields of view. (**F**) Map of x/y-localisations determined by point-spread function fitting of single AF647-PFO molecules transiently binding to the same single liposome as in A-C. (**G**) Super-resolved image showing liposome outlines reconstructed from single AF647-PFO localisations. The reconstruction inside the box corresponds to the localisations shown in C. (**H**) Diffraction-limited TIRF image of the AF488 content dye of the same area as shown in D. An overlay of a larger region of the field of

*Figure 2 continued on next page*

*Figure 2 continued*

view in both channels is shown in *Figure 2—figure supplement 1*. (**I–K**) Analysis of AF647-PFO monomer binding. (**I**) Distribution of dwell times in the bound state (blue line) extracted from the trace shown in A. An exponential fit of the curve is shown in orange. The exponent of this fit (0.0025 frames$^{-1}$) provides the monomer binding rate for this liposome. (**J**) Distributions of binding rates from all liposomes in the field of view measured at different AF647-PFO concentrations. (**K**) Monomer binding rates increase linearly with AF647-PFO concentration. Each data point represents the median binding rate determined from the corresponding distribution recorded in an independent TIRF image stack (3 experiments at 2.5 pM, 4 experiments per concentration between 5–15 pM). The orange line represents a linear fit of the data, whereby the slope of the line provides an estimate of the monomer binding rate constant. (**L–N**) Analysis of AF647-PFO monomer unbinding. (**L**) Distribution of waiting times between molecules (blue line) extracted from the trace shown in A. An exponential fit of the curve is shown in orange. The exponent of this fit (0.086 frames$^{-1}$) is corrected for photobleaching (0.0517 frames$^{-1}$) and then taken as the monomer unbinding rate (0.0343 frames$^{-1}$, 0.6 s$^{-1}$) for this liposome. The photobleaching correction is validated in *Figure 2—figure supplement 3*. (**M**) Distribution of unbinding rates from all liposomes in the field of view measured at different AF647-PFO concentrations. (**N**) Monomer unbinding is independent of concentration. Each data point represents the median unbinding rate determined from the corresponding distribution recorded in the same 23 experiments as in K. The orange line represents the median of all experiments.

The online version of this article includes the following video and figure supplement(s) for figure 2:

**Figure supplement 1.** Single-molecule AF647-PFO binding events co-localise with Alexa Fluor 488-loaded liposomes.

**Figure supplement 2.** Liposomes retain their content dye during single-molecule PFO binding experiments.

**Figure supplement 3.** Validation of photobleaching correction of single-molecule unbinding rates.

**Figure supplement 4.** In silico modelling of early-stage PFO assembly and conversion to irreversible dimer (nucleation).

**Figure 2—video 1.** Morphing movie of the two dimer conformational states predicted by AlphaFold.

https://elifesciences.org/articles/74901/figures#fig2video1

partial or multi-step signal loss (*Figure 1D and F*). The remainder (~10%) showed no or little signal loss (*Figure 1E and F*), suggesting that these were not permeabilised despite AF647-PFO binding to many of these liposomes (further examples shown in *Figure 1—figure supplement 3B*). At 50 pM AF647-PFO, single-step pore formation was less efficient (<10%), and at even lower AF647-PFO concentrations, dye release was no longer observed (*Figure 2—figure supplement 2*). We tentatively attribute the different release profiles to heterogeneity in the liposome preparations. For example, aggregated liposomes and multilamellar liposomes would be expected to give rise to traces with multiple steps or partial release of dye. Alternatively, multiple dye release steps could arise from transient pore opening events with a lifetime in the range of 0.1–1 ms (*Figure 1—figure supplement 4*).

Single-step traces were identified for further analysis and initial inspection revealed that the time for nucleation of a growing PFO oligomer and the time for poration varied between liposomes, as did the number of subunits at the time of pore opening. These processes are analysed in more detail below. As expected, the kinetics of liposome poration decreased with decreasing AF647-PFO concentration (*Figure 1G*). At high concentrations (500 pM, *Figure 1G*, light blue) single-step dye release was complete within ~20 min, while at low concentrations (50 pM, *Figure 1G*, dark blue) this process takes ~2 hr.

## Characterisation of PFO monomer binding to liposomes

The first step of PFO pore formation involves the binding of PFO monomers to the membrane (*Figure 1A*, left). To measure the kinetics of this process, we imaged the interactions of AF647-PFO molecules with liposomes at high temporal resolution (~17.5 frames/s) for a period of 7 hr at room temperature at very low concentrations (2.5–20 pM), where the chance of two monomers binding to a liposome at the same time to dimerise becomes negligible. Under these conditions, we observed transient binding of single AF647-PFO molecules appearing as sporadic fluorescent spots. We then used a software developed for single-molecule localisation microscopy (*Schnitzbauer et al., 2017*) to detect and track AF647-PFO spots appearing and disappearing at the locations of individual liposomes, resulting in a trace at each location with typically over 1000 binding events. A representative single-liposome trace recorded in the presence of 10 pM AF647-PFO shows that the rate of binding events remained constant over the entire 7-hr experiment (*Figure 2A*), while zoomed-in views show sporadic binding intensity spikes (*Figure 2B*) corresponding to a AF647-PFO signal persisting for several frames (*Figure 2C*). The intensity distribution of all binding events on all liposomes overlaid completely with the intensity distribution of the same batch of AF647-PFO molecules sparsely immobilised directly on a glass coverslip (*Figure 2D*) indicating that a single species associated with the liposomes. Single-molecule photobleaching of the molecules adhered to a glass coverslip showed

that most molecules (84%) bleached in a single step (*Figure 2E*). Taken together, these observations suggest that the AF647-PFO species transiently binding to liposomes are likely to be monomers.

The scatter plot of x/y-coordinates for all molecules detected in *Figure 2A–C* forms a ring-like distribution, as expected for the z-projection of AF647-PFO molecules binding to the membrane of a spherical liposome (*Figure 2F*). The x/y-coordinate maps for all liposomes were further used to reconstruct a super-resolved image, which revealed the outlines of liposomes bound to the cover-slip (*Figure 2G*). Interestingly, many liposomes displayed bright spots in the reconstruction images suggesting that liposome membranes may have a spatially inhomogeneous affinity for binding PFO, but the underlying mechanism for this observation remains unclear. Importantly, the super-resolved AF647-PFO structures colocalised with the diffraction-limited signals in the AF488 content dye channel, confirming that binding occurred on liposomes (*Figure 2G/H* and *Figure 2—figure supplement 1*).

Next, we determined the rates of AF647-PFO monomer binding and unbinding from each binding trace collected at a single liposome. First, we measured the time intervals between peaks in the trace (an example of one such $\Delta t_{dark}$ is highlighted in *Figure 2B*) to determine the distribution of waiting times before binding of the next molecule (*Figure 2I*). An exponential fit of this distribution provided the AF647-PFO monomer binding rate (0.043 $s^{-1}$ for the example liposome shown in *Figure 2I*). Similarly, we measured the duration of each peak (an example of one such $\Delta t_{bright}$ is highlighted in *Figure 2C*) to determine the time distribution of the bound state (*Figure 2L*). An exponential fit of this distri-bution then provided the AF647-PFO monomer unbinding rate after correcting for photobleaching (*Figure 2—figure supplement 3*). We repeated this analysis for all liposomes in the field of view, revealing a wide distribution of binding and unbinding rates on different liposomes (*Figure 2J and M*, respectively), possibly due to inhomogeneity in binding affinities of different structural states and/or local lipid composition. Analysis of binding experiments at a range of AF647-PFO concentrations showed that the binding rate distributions shifted to higher values with concentration (*Figure 2J*), whereby the median monomer binding rate increased linearly with concentration (*Figure 2K*). This analysis allowed us to obtain the monomer binding rate constant ($B$=0.9 ± 0.23 $nM^{-1}$ $s^{-1}$) which corre-sponds to 7.2±1.8 $nM^{-1}$ $s^{-1}$ $\mu m^{-2}$ when taking the surface area of liposomes into account (calculated as the surface area of a 200 nm sphere). As expected, the unbinding rate distributions (*Figure 2M*) and the median unbinding rates obtained from these distributions were independent of concentra-tion (*Figure 2N*), yielding a value for the unbinding rate of $U$=2.27 ± 0.81 $s^{-1}$. Additional experiments confirmed that the unbinding rate was independent of laser power (*Figure 2—figure supplement 3*). See *Table 1* for a summary of all parameters measured in this work.

Taken together, our observations show that single AF647-PFO molecules rapidly cycle between the solution and the membrane, whereby the membrane-bound state is short-lived (half-life of 0.3 s). As expected, these interactions did not lead to pore formation, as liposomes retained their content dye for the duration of the experiments (*Figure 2—figure supplement 2*).

**Table 1.** PFO pore formation kinetic and thermodynamic parameter values.
All errors are the standard deviation of independent measurements (see Appendix).

| Parameter | Symbol | Value | Units |
|---|---|---|---|
| Monomer binding rate constant | $B$ | 0.9±0.23 | $nM^{-1}$ $s^{-1}$ |
| Monomer unbinding rate | $U$ | 2.27±0.81 | $s^{-1}$ |
| Dimer formation rate constant | $D_f$ | 0.16±0.19 | $nM^{-1}$ $s^{-1}$ |
| Dimer dissociation rate | $D_r$ | (8.6±3.4) x$10^{-4}$ | $s^{-1}$ |
| Oligomerisation rate constant | $P$ | 0.23±0.028 | $nM^{-1}$ $s^{-1}$ |
| Insertion rate (at 22 °C) | $I_0$ | (5.0±2.4) x$10^{-4}$ | $s^{-1}$ |
| | $I_{g0}$ | (4.0±1.6) x$10^{-5}$ | $s^{-1}$ |
| | $I_{gc}$ | (4.25±1.45) x$10^{-4}$ | $nM^{-1}$ $s^{-1}$ |
| Activation energy for insertion | $E_a$ | (1.8±0.63)x$10^{-19}$ | J |
| | | 26.2±9.12 | kcal/mol |

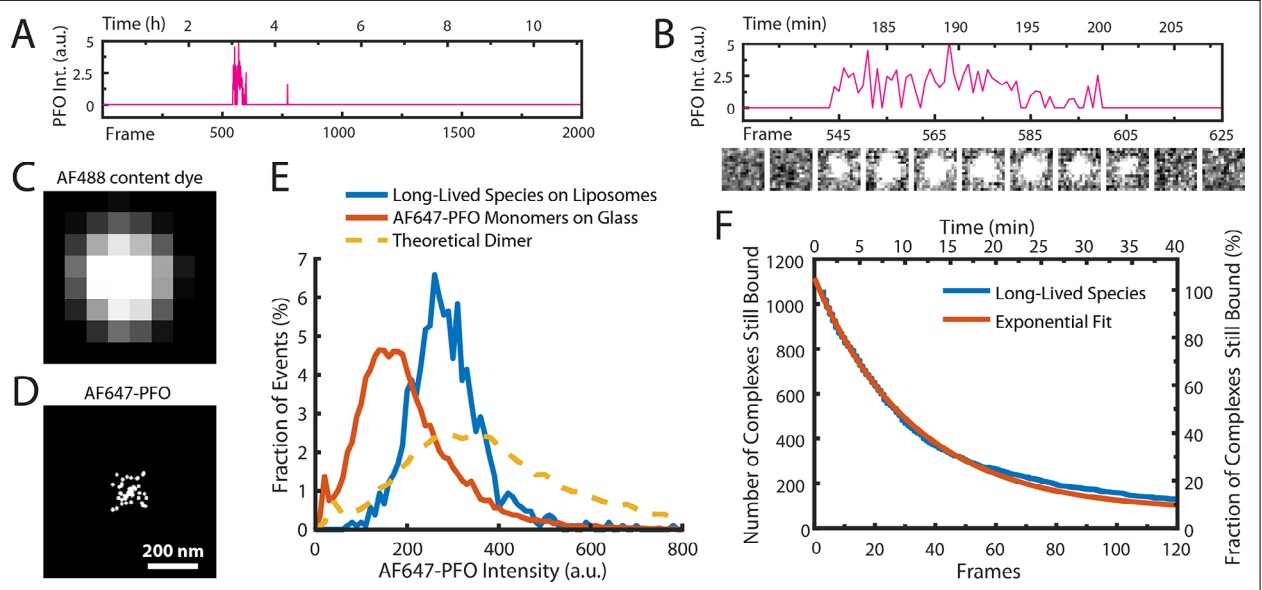

**Figure 3.** AF647-PFO dimers can stably bind to liposomes. (**A,B**) Trace containing a long-lived state detected at the location of a single liposome from a time series (2000 frames at 3 frames per minute) acquired in the presence of 10 pM AF647-PFO. Under these conditions, detection of short-lived monomers is rare. (**B**) Expanded view of the peak in A, with the corresponding snapshots (1.76 μm × 1.76 μm, average of 5 consecutive frames) shown below. (**C,D**) Diffraction-limited TIRF image of the AF488 content dye (**C**) and corresponding map of all x/y-localisations over time of the single AF647-PFO species (**D**) that remains bound to the liposome for the duration of the peak in panel A. The scatter of localisations reflects the diffusion of the species on the liposome surface. (**E**) Intensity distributions of long-lived states on liposomes (blue) and of immobilised AF647-PFO monomers on a glass (orange). The yellow dashed line is the theoretical distribution for AF647-PFO dimers calculated from the monomer intensity distribution. (**F**) Distribution of dwell times in the long-lived bound state (blue line) extracted from 1063 events acquired in five experiments. An exponential fit with a decay constant of 0.03 frame$^{-1}$ is shown in orange. After correction for photobleaching (0.0133 frame$^{-1}$), this analysis gives a dimer disappearance rate of $D_{off}$ = 0.052 min$^{-1}$.

The online version of this article includes the following figure supplement(s) for figure 3:

**Figure supplement 1.** Comparison of state lifetimes for experiments run at 3 frames per minute and 0.5 frames per minute.

## PFO dimerisation on the membrane produces a metastable complex

The fast frame rate and high laser power required to image the binding of single AF647-PFO molecules to liposomes (*Figure 2*) made it impossible to observe longer-lived species as they would be rapidly photobleached. TIRF binding experiments with a much slower frame rate (3 frames per minute) and lower laser power allowed us to detect rare long-lived species (1063 events from ~7500 liposomes viewed over 11 hr) as illustrated by the trace with an AF647-PFO signal that persisted for ~20 min (*Figure 3A/B*). This signal colocalised with an AF488-loaded liposome (*Figure 3C/D*), consistent with binding to the membrane.

To determine the stoichiometry of AF647-PFO in the long-lived state, we measured the average intensity over time of each long-lived liposome-bound signal. The resulting intensity distribution showed that the species was almost twice as bright as monomeric AF647-PFO on glass and agreed more closely with the predicted intensity distribution (*Mutch et al., 2007*) of a dimer (*Figure 3E*). To determine the dissociation kinetics of this species, we measured the duration of all long-lived signals. The resulting distribution decayed exponentially (*Figure 3F*), suggesting that signal disappearance is governed by a single rate-limiting step with a rate of $D_r$ = 8.6 ± 3.4 x 10$^{-4}$ s$^{-1}$ after correction for photobleaching (see *Table 1* for parameter summary). Finally, we tested whether even longer-lived species could be detected by further decreasing the frame rate (0.5 frames per minute), but only recovered the same dimer species described above (*Figure 3—figure supplement 1*). Changing the frame rate by a factor of 6 did not change the calculated dimer dissociation rate (*Figure 3—figure supplement 1*), strongly suggesting that the release of the dimer from the membrane is not the result of photobleaching artifacts.

Taken together, this analysis shows that at low concentrations, the interactions of AF647-PFO on liposomes very rarely lead to the formation of a metastable dimer on the membrane that persist with

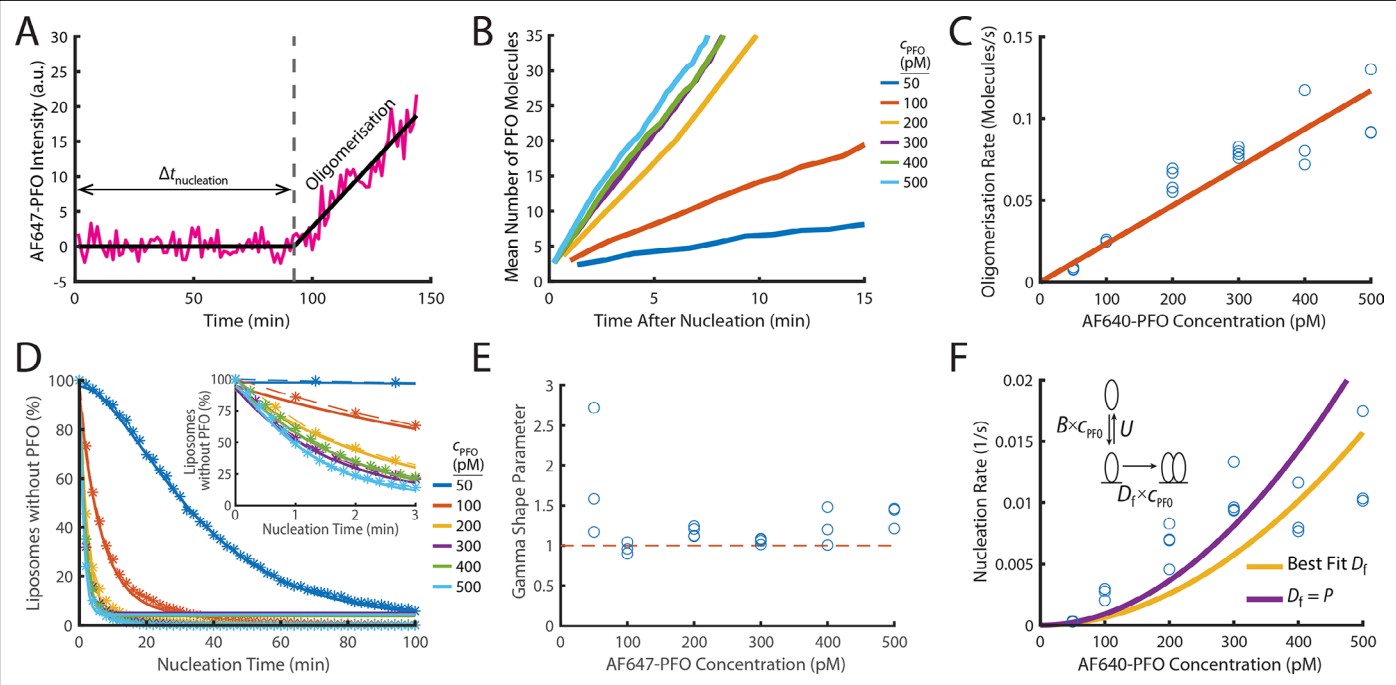

**Figure 4.** PFO nucleation and oligomerisation. (**A**) AF647-PFO intensity trace of the early stages of PFO assembly showing fluctuations around the baseline (before nucleation) and signal increase due to oligomerisation (after nucleation). The slope of the linear fit line provides the oligomerisation rate. The nucleation time ($\Delta t_{nucleation}$) is defined at the time where the oligomerisation fit line intersects the baseline. (**B**) The mean of all traces after they have been aligned to their time of nucleation displays a linear increase in intensity resulting from oligomerisation. (**C**) Oligomerisation rate as a function of AF647-PFO concentration; each data point represents the median oligomerisation rate determined in an independent PFO pore formation experiment (3 experiments for 50 pM, 100 pM, 400 pM, 500 pM; 4 experiments for 200 pM, 300 pM). The slope of the linear fit (orange line) provides the oligomerisation rate constant p=0.23 nM⁻¹ s⁻¹. (**D**) Experimental nucleation time distributions measured at different concentrations are represented by dashed lines with stars. Each distribution is an average of at least three experiments. Gamma distributions fitted to the experimental data are represented by solid lines. Inset shows a zoom in of the first 3 minutes (**E**) Gamma shape parameter vs. AF647-PFO concentration determined from the fits of the nucleation time distributions in (**D**). The gamma distribution fit to the nucleation times for 100–500 pM yielded a shape parameter of approximately 1 (mean of 1.09), consistent with a single rate-limiting step. A shape parameter of 1 is the equivalent to an exponential fit. In contrast, fitting the pooled 50 pM data required a shape parameter of 2.02, suggesting an additional step becomes rate-limiting at this concentration. (**F**) Nucleation rate as a function of AF647-PFO concentration. The value for $D_f$ was obtained by fitting ("Best fit", yellow line), giving a value of 0.16 nM⁻¹ s⁻¹. Alternatively, the value for $D_f$ was assumed to be identical to the oligomerisation rate constant $P$=0.16 nM⁻¹ s⁻¹ (purple line). Inset: Kinetic model for predicting the concentration dependence of the nucleation rate, where $B$ is the monomer binding rate constant, $U$ is the monomer unbinding rate, $D_f$ is the dimerisation rate constant and $c_{PFO}$ is the PFO concentration. The values for $B$ and $U$ were obtained from the single-molecule binding experiments in *Figure 2*.

The online version of this article includes the following figure supplement(s) for figure 4:

**Figure supplement 1.** Exponential fits of nucleation rates.

**Figure supplement 2.** Comparison of dimer release models.

**Figure supplement 3.** Direct measurement of PFO monomer addition using step fitting.

a half-life of 13 min (corresponding to a mean lifetime of 20 min), that is three orders of magnitude longer than the monomer. We conclude that dimerisation on the membrane promotes strong PFO–PFO and PFO–membrane interactions. From our data we cannot distinguish whether the dimer disappears by unbinding from the membrane, or whether it dissociates to monomers that subsequently unbind from the membrane. We also note that the half-life of the dimer state is sufficiently long such that it is essentially irreversible on the time scale of PFO assembly at concentrations of ≥100 pM, where the entire pore formation process takes on average less than 20 min (*Figure 1G*).

## PFO dimerisation on the membrane nucleates a stably growing oligomer

At higher concentrations (≥25 pM), the transient interactions of PFO eventually lead to nucleation of a membrane-bound oligomer that continues to grow (*Figure 1C* and *Figure 4A–C*). The nucleation times determined from AF647 traces recorded at concentrations between 100–500 pM followed single exponential distributions (*Figure 4D/E*, gamma shape parameters ≈1), suggesting that nucleation is governed by a single rate-limiting step. The nucleation times at 50 pM were better described by a gamma distribution with a shape factor of ~2 (*Figure 4D/E*), suggesting that at very low PFO concentrations close to the threshold where nucleation is no longer observed, nucleation is also governed by an additional rate-limiting step. The nucleation rate determined from the exponential fit of the nucleation time distribution increased with AF647-PFO concentration (*Figure 4F*, *Figure 4—figure supplement 1*); at higher concentrations it becomes more probable for a second PFO molecule to bind to the membrane before the first one has fallen off.

To determine the post-nucleation kinetics of AF647-PFO oligomerisation, we generated the mean AF647-PFO intensity trace by aligning all traces recorded at single liposomes at the time of nucleation. The mean traces obtained for a range of concentrations (*Figure 4B*) show that oligomerisation occurs at a constant rate (all lines are linear) that increases with concentration. To further quantify the dependence of oligomerisation on concentration, we fitted the steady increase in fluorescence intensity after nucleation for each liposome with a linear function, whereby the slope provided the oligomerisation rate. As expected, the median oligomerisation rate for all liposomes increased linearly with AF647-PFO concentration (*Figure 4C*), providing an oligomerisation rate constant of p=0.23 ± 0.028 nM$^{-1}$ s$^{-1}$. This oligomerisation rate is essentially the product of the rate of PFO monomer binding to the membrane and the probability of finding and joining the growing oligomer. The ratio between the oligomerisation rate constant (0.23 nM$^{-1}$ s$^{-1}$) to the single molecule binding rate constant (0.9 nM$^{-1}$ s$^{-1}$) reveals that 25% of the time, a PFO monomer binding to the membrane will end up joining the oligomer (see *Table 1* for parameter summary). Finally, we validated the oligomerisation rate by directly observing the successive arrival of AF647-PFO monomers using single molecule imaging conditions, which yielded a value of 0.15±0.04 nM$^{-1}$ s$^{-1}$ (*Figure 4—figure supplement 3*). Since this approach does not rely on fluorescence intensity, the good agreement between the two independent measurements also validates the intensity-based method to determine the number of molecules in oligomers used in this work.

The oligomer disassembly rate (given by the Y-intercept of the fit line in *Figure 4C*) is effectively zero suggesting that PFO oligomers are stable on the membrane and do not dissociate. We conclude that PFO oligomerisation is essentially irreversible on the time scale of pore formation. The high stability of PFO oligomers on the membrane is also observed after removal of AF647-PFO from solution (*Figure 5—figure supplement 1C*), as discussed below, and is not surprising given the long mean lifetime of the dimer on the membrane (~20 min).

On the basis of the high dimer stability, we reasoned that the dimer represents the stable nucleus for oligomerisation, that is the rate-limiting step for nucleation is waiting for two monomers to be bound to the membrane at the same time and form a dimer. We used the kinetic model shown in the inset of *Figure 4F* to calculate theoretical nucleation rates, whereby firstly, membrane binding is governed by the binding and unbinding kinetics of monomers (*Figure 2*) and secondly, dimerisation on the membrane is governed by a dimerisation rate constant, $D_f$. We fitted the model to the experimental concentration dependence of nucleation rates (*Figure 4F*) with $D_f$ as the only free parameter to obtain an estimate of $D_f$ = 0.16 ± 0.19 nM$^{-1}$ s$^{-1}$, similar to the value for the oligomerisation rate constant obtained above (p=0.23 ± 0.028 nM$^{-1}$ s$^{-1}$,*Table 1*). Thus, dimerisation is kinetically similar to oligomerisation, consistent with the dimer being the first stable species on the membrane. Accordingly, the predicted nucleation rates when using the value of *P* to parametrise dimerisation in the model were also in reasonable agreement with the experimental data. This agreement is remarkable, given that parameter values for the monomer binding/unbinding and oligomerisation kinetics were obtained using different experimental conditions and are sufficient to predict nucleation rates.

Next, we extended the kinetic model to account for the finite dimer stability measured in *Figure 3*. Dimer dissociation may occur via one of two alternative pathways, that is unbind from the membrane or fall apart into monomers. Fits of these extended models to the experimental concentration dependence of nucleation rates were essentially indistinguishable from the simple model described above

(*Figure 4—figure supplement 2*). This is because the dimer dissociation process is so slow, that it has virtually no impact on the outcome of nucleation in the concentration range investigated here, regardless of the pathway.

## PFO membrane insertion kinetics increase with the number of subunits in the oligomer

Dual-colour pore formation traces showed that membrane pore opening (dye release) occurred after onset of the AF647-PFO signal increase (*Figure 1—figure supplement 3*). Analysis of the number of subunits at the time of dye release showed that low stoichiometry PFO oligomers insert into the membrane to form pores before they form full rings. While the fluorescence data do not provide structural information, we interpret these low stoichiometry PFO oligomers as arcs since wild type PFO assembly intermediates appear as arc-shaped structures in AFM images (*Czajkowsky et al., 2004*). The pore opening kinetics of continuously growing arcs is complicated and analysed in detail below. To first measure (arc) pore opening kinetics in the absence of continuing oligomerisation, we devised a wash-out assay, in which liposomes are exposed to AF647-PFO for a limited time to allow formation of membrane-bound arcs. The AF647-PFO was then washed out of the microfluidics channel using a

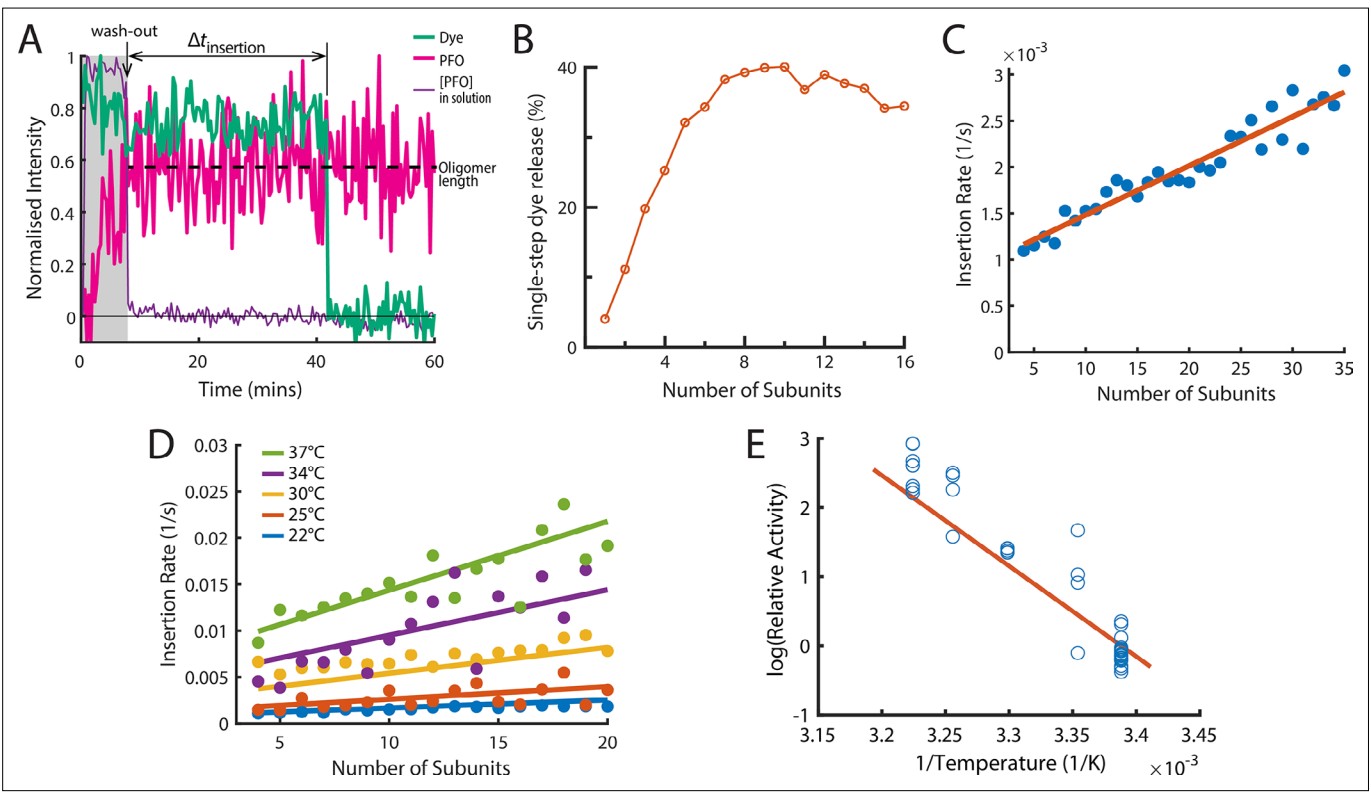

**Figure 5.** The energetics of PFO oligomer membrane insertion depends on the number of subunits in the prepore. (**A**) Example trace of the AF647-PFO washout experiment. The purple line represents the background intensity of AF647-PFO in solution; the blue-green line represents the intensity of the AF488 content dye marker; the magenta line represents the AF647-PFO intensity on the liposome. The insertion time is determined as the waiting time between loss of the background signal (wash-out) and loss of content dye signal (pore opening). The AF647-PFO intensity at the time of wash-out (dotted line) was used to determine the number of subunits in the prepore. (**B**) Pore opening efficiency (determined as the fraction of liposomes that release the content dye in a single step) as a function of the number of subunits. (**C**) Insertion rate (determined from the exponential fit of the corresponding insertion time distribution) as a function of the number of subunits with a line of best fit. (**D**) Insertion rate as a function of the number of subunits determined at different temperatures. The fit lines were obtained by a global fit of the data (weighted on the basis of the number of liposome observations), whereby the only free parameter of the fit is proportional to the activation energy. (**E**) Arrhenius plot of the relative activity of insertion (blue data points) obtained from the slopes of local fits of the data in D. The slope of the orange line is obtained from the parameter value obtained from the global fit in D. Temperature data consists of 18 experiments at 22 °C, 4 experiments at 25 °C, 5 experiments at 30 °C, 4 experiments at 34 °C, and 8 experiments at 37 °C.

The online version of this article includes the following figure supplement(s) for figure 5:

**Figure supplement 1.** Number of subunits in the PFO oligomer during washout.

buffer at a time when the majority of liposomes still retained their content dye (i.e. the pore had not yet formed).

After wash-out, we observed that these dye-loaded liposomes showed a wide distribution of AF647-PFO oligomer sizes between 2 and 40 subunits (*Figure 5—figure supplement 1A*), as expected for a stochastic nucleation and growth process. We then continued to image the liposomes by TIRF microscopy to detect the dye release as a read-out for pore formation. A typical example trace recorded at a single liposome is shown in *Figure 5A*. Initially, AF647-PFO nucleated a growing arc (signal appearance and increase) on the membrane, while the content dye signal remained high. After wash-out, the AF647-PFO signal stayed constant for the remainder of the experiment, confirming that membrane-bound arcs did not release PFO subunits (see also *Figure 5—figure supplement 1C*). After a waiting time, the content dye signal disappeared, which we interpret as the insertion of the PFO arc into the membrane and concomitant opening of the membrane pore. Thus, we defined the time period between wash-out and dye release as the insertion time. Since our experimental read-out (dye release) reports the opening of the membrane pore, this definition assumes that the rearrangement of lipids required to open the semi-toroidal membrane pore is fast compared to time required to wait for PFO insertion.

When applied to all liposomes, this analysis allowed us to determine the efficiency and kinetics of pore formation for arcs containing a defined number of subunits (oligomer size). First, only a low fraction of liposomes with AF647-PFO dimers (~12%) or trimers (~20%) released their content dye, but pore formation efficiency increased sharply with oligomer size, reaching half-maximal efficiency for tetramers and becoming size-independent for arcs containing at least six subunits (*Figure 5B*). We conclude that arcs with at least four subunits could efficiently insert into the membrane resulting in the opening of a transmembrane pore, as observed in molecular dynamics simulations of a membrane-inserted CDC pentamer (*Vögele et al., 2019*). Second, we extracted the insertion time distributions for each oligomer size. These distributions could be described with a single exponential function to yield the oligomer-size specific insertion rate (assumed to be the rate-limiting step for pore opening). Surprisingly, the insertion rate increased linearly with the number of subunits in the oligomer (*Figure 5C*), from species corresponding to tetramers to full rings. Taken together, these observations suggest a stochastic trigger for the prepore to pore transition as discussed in more detail below.

## Membrane insertion represents the main energy barrier for pore formation

We repeated the wash-out experiments at a range of temperatures to determine the activation energy of PFO oligomer insertion. First, we extracted dependence of insertion rates on the number of subunits in the oligomer between 22 and 37°C (*Figure 5D*). Fitting of the relative activities at each temperature provided an activation energy of $26.2 \pm 9.12$ kcal mol$^{-1}$ for the transition from the prepore to the open pore state; this analysis is shown as an Arrhenius plot in *Figure 5E*. For comparison, the activation energy determined from ensemble measurements for the entire pore formation pathway are similar ($23.9 \pm 1.3$ *Wade et al., 2015a* or $28 \pm 1.9$ kcal mol$^{-1}$ *Wade et al., 2019*). We conclude that the insertion event is the dominant energy barrier during pore formation.

## Pore insertion kinetics for continuously growing arcs

The dual-colour pore formation experiments showed that even in the continued presence of AF647-PFO in solution, arcs instead of full rings (as judged from the number of subunits determined from the intensity of the AF647-PFO signal) were the most common species to perforate the liposome membrane. To obtain an estimate of the insertion rate as a function of the number of subunits in the case of a continuously growing arc, we defined a normalised rate as follows. We divided the total number of insertion events for each given oligomer size (number of subunits) by the cumulative observed time across all detected instances of AF647-PFO oligomers of that size. As observed in the wash-out experiments above, the insertion rate depended linearly on the number of subunits in the oligomer at AF647-PFO concentrations between 50 and 500 pM (*Figure 6A*). Surprisingly, the insertion rate also increased linearly with the AF647-PFO concentration in solution (*Figure 6A* and *Figure 6—figure supplement 1*) via an unidentified mechanism.

To predict the number of subunits in the oligomer and the kinetics of insertion, we developed a kinetics model (*Figure 6B*, explained in detail in the Appendix) that is entirely parameterised by

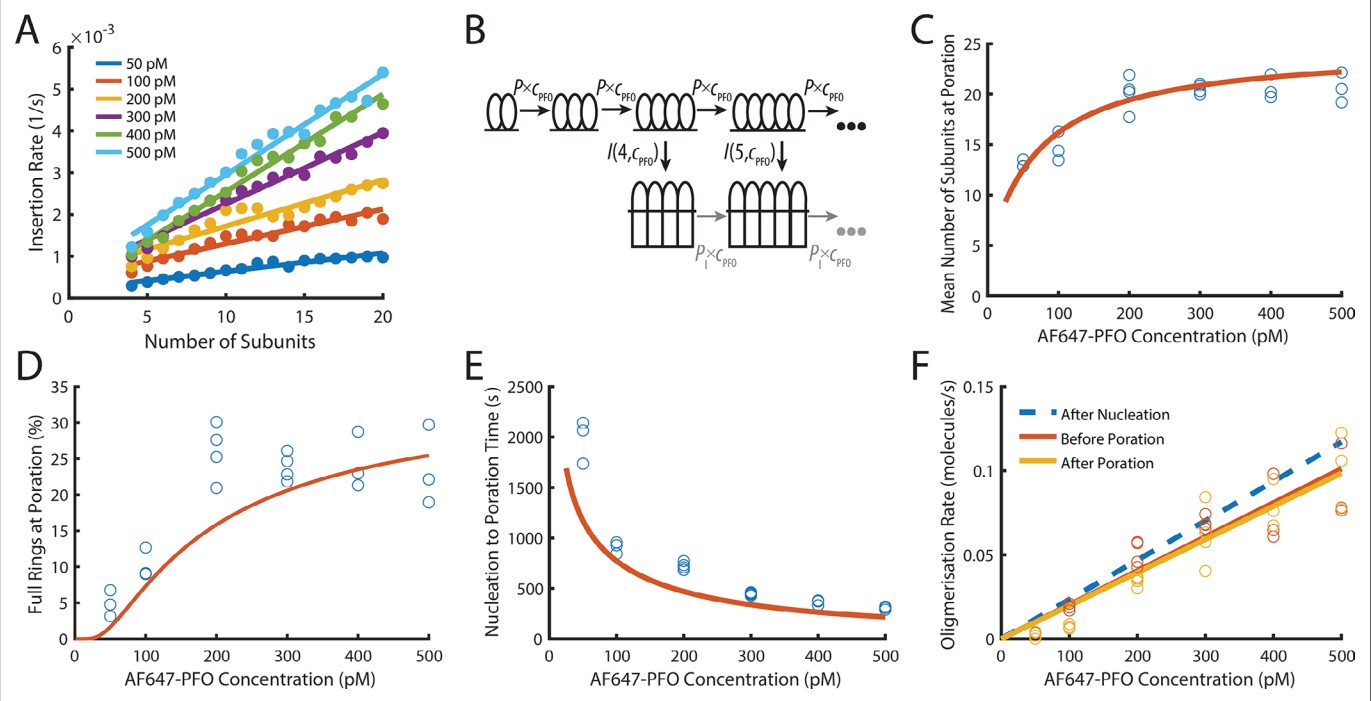

**Figure 6.** PFO insertion kinetics during continuous growth. (**A**) The insertion rate of continuously growing PFO arcs depends on the number of subunits in the arc (n) and the concentration of AF647-PFO in solution ($c_{pfo}$) and is given by: $I(n,c_{pfo}) = I_0 + n(I_{g0} + c_{pfo} \times I_{gc})$ where $I_0$, $I_{g0}$ and $I_{gc}$ are the kinetic parameters for insertion (See **Figure 6—figure supplement 1**). (**B**) Kinetic model used to predict the number of subunits in the prepore and the insertion kinetics. Dimerisation is the committed step for continued oligomerisation. Arcs with at least four subunits can insert into the membrane (or continue to grow), whereby the insertion rate increases with the number of subunits in the prepore (**A**). The parameter values were obtained from experiment and are shown in **Table 1**. (**C**) Mean number of subunits at the time of poration as a function of AF647-PFO concentration. Each data point is the mean from an independent experiment. The orange line shows the prediction from the model in B. (**D**) Fraction of complete rings formed at the time of poration. (**E**) Mean time from nucleation to poration as a function of AF647-PFO concentration. The orange line shows the prediction from the model in B. (**F**) Oligomerisation kinetics are the same before (yellow) and after (orange) poration. The fit line of the oligomerisation kinetics measured after nucleation (**Figure 3D**) is shown as a dashed blue line. The graphs in **A, B, C, D, E, F** are derived from the same data as **Figure 3** (3 experiments for 50 pM, 100 pM, 400 pM, 500 pM; 4 experiments for 200 pM, 300 pM).

The online version of this article includes the following figure supplement(s) for figure 6:

**Figure supplement 1.** Fitting Concentration Dependence of Insertion.

**Figure supplement 2.** Arc size distribution at the time of Insertion.

**Figure supplement 3.** Hypothetical models of PFO pore growth for an inserted arc pore.

**Figure supplement 4.** Negative-staining electron microscopy of PFO on POPC/cholesterol liposomes.

**Figure supplement 5.** Probability of nucleating a second PFO oligomer on the liposome before the first oligomer has inserted into the membrane.

the experimentally determined rates for oligomerisation and insertion. In this model, the assembly process starts from a stable dimer nucleus. Below the minimum number of subunits in the oligomer required for efficient insertion and pore opening (n<4), the oligomer grows by monomer addition. Once the oligomer reaches n=4, it can either grow or insert, whereby the probability of growth versus insertion is determined by the respective rates for these processes.

Next, we validated the model by comparing predictions to experimentally determined distributions of intermediates detectable in the dual colour pore formation experiments. The size of oligomers inserting into the membrane, determined as the number of subunits from the AF647-PFO signal at the time of dye release, was broadly distributed at all AF647-PFO concentrations (**Figure 6—figure supplement 2A**), whereby the upper end of the distributions represented the level expected for full rings (~35 subunits **Czajkowsky et al., 2004**; **Dang et al., 2005**). The mean number of subunits in the pore at insertion (**Figure 6C**) increased with concentration from fewer than 15 subunits at 50 pM to 20 subunits at 500 pM. Similarly, the fraction of oligomers inserting at a number of subunits consistent with a complete ring increased with concentration (**Figure 6D**) but remained below 30% even

at the highest concentrations tested here. This outcome reflects that the oligomerisation rate was on average not fast enough to complete the full ring before arc insertion occurred. The kinetic model faithfully reproduced the concentration dependence of size distributions (*Figure 6—figure supplement 2*), mean subunits at insertion (*Figure 6C*, orange line) and predicted the fraction of complete rings (*Figure 6D*, orange line) at insertion within a factor of two.

TIRF imaging at higher laser power and temporal resolution to detect the addition of single PFO monomers to oligomers yielded similar values to the ones described above for the the number of PFO monomers in the oligomer at the time of membrane insertion and the fraction of oligomers that reach the requisite number of subunits to insert as closed rings (*Figure 4—figure supplement 3E,F*). This analysis provides an intensity-independent validation of these parameters.

From the single-liposome traces we were also able to measure the time required from nucleation to opening of the membrane pore (*Figure 6E*). On average, this time decreased with AF647-PFO concentration, as expected given that arcs grow more quickly at higher concentration and longer arcs insert more quickly. This trend was also correctly predicted by the model (*Figure 6E*, orange line). Overall, the excellent agreement between data and model predictions supports the dependence of insertion on the number of PFO subunits in the oligomer and concentration.

## PFO oligomers continue to grow after insertion

Next, we analysed the AF647-PFO signal before and after the transition of the arc in the prepore state to the pore state had occurred (*Figure 6F*). It has previously been shown that arcs of the related CDC suilysin do not continue to grow after the transition from the prepore to the pore state (*Leung et al., 2014*). Surprisingly, the AF647-PFO oligomerisation rate was the same immediately after poration (dye release) (*Figure 6F*, yellow symbols) as it was just before poration (*Figure 6F*, orange symbols), that is the AF647-PFO signal increase did not stop (or pause) before or after insertion. Since the nucleation of a new structure is a slow process that would lead to a pause in the AF647-PFO signal increase, we interpret this observation as the continued addition of monomers (that bind from solution to the membrane) to the arc pore. An alternative explanation, that is the simultaneous growth of a second oligomer on the liposome, is unlikely for the following reasons: The nucleation of a second oligomer before insertion is unlikely since the first oligomer acts as a sink for monomers, further slowing an already slow nucleation step. Also, the appearance of additional oligomers would be apparent from an increase in the oligomerisation rate, which is not observed in our data (*Figure 6F*, orange symbols). This interpretation of nucleating a single oligomer per liposome is also consistent with negative staining EM images of PFO structures which appear as single complete rings, despite being assembled at relatively higher (nanomolar) concentrations (*Figure 6—figure supplement 4*). Finally, we calculated the upper limit for the probability of nucleating a second dimer on a liposome before insertion of the growing PFO oligomer (*Figure 6—figure supplement 5*). This analysis showed that the majority of liposomes are predicted to contain a single oligomer at the time of dye release at concentrations ≤100 nM, but nucleation of a new dimer become probable at higher concentrations (note that this dimer may not form an independently growing oligomer but could join the first oligomer).

## Pore opening coincides with a drop in PFO oligomer movement on the membrane

Membrane insertion leads to a drastic reduction of the diffusion rate of CDC complexes on the surface of flat membranes (*Senior et al., 2022*; *Leung et al., 2014*). To determine whether the same effect could be observed on liposomes, we imaged PFO assembly and pore formation assay with high laser intensity to accurately localise the position (projected onto the x/y-plane) of growing PFO oligomers on liposomes over time with sub-pixel resolution. The example trace in *Figure 7A* shows that the frame-to-frame movements of the AF647-PFO signal initially fluctuated between 0 and 300 nm (*Figure 7A*). Pore opening (detected by dye release) led to a pronounced drop in movement (*Figure 7A* after 7.5 min). This reduction in movement is also evident in the map of all x/y-localisations. Localisations appeared as a diffuse point cloud with approximately the same size as the liposome (*Figure 7B* top left versus bottom right), and formed a tight focus upon dye release.

Experiments conducted at 100–500 pM AF647-PFO showed that oligomers on 85% of liposomes showed a reduction of ~41% in movement across the entire concentration range, which occurred in a

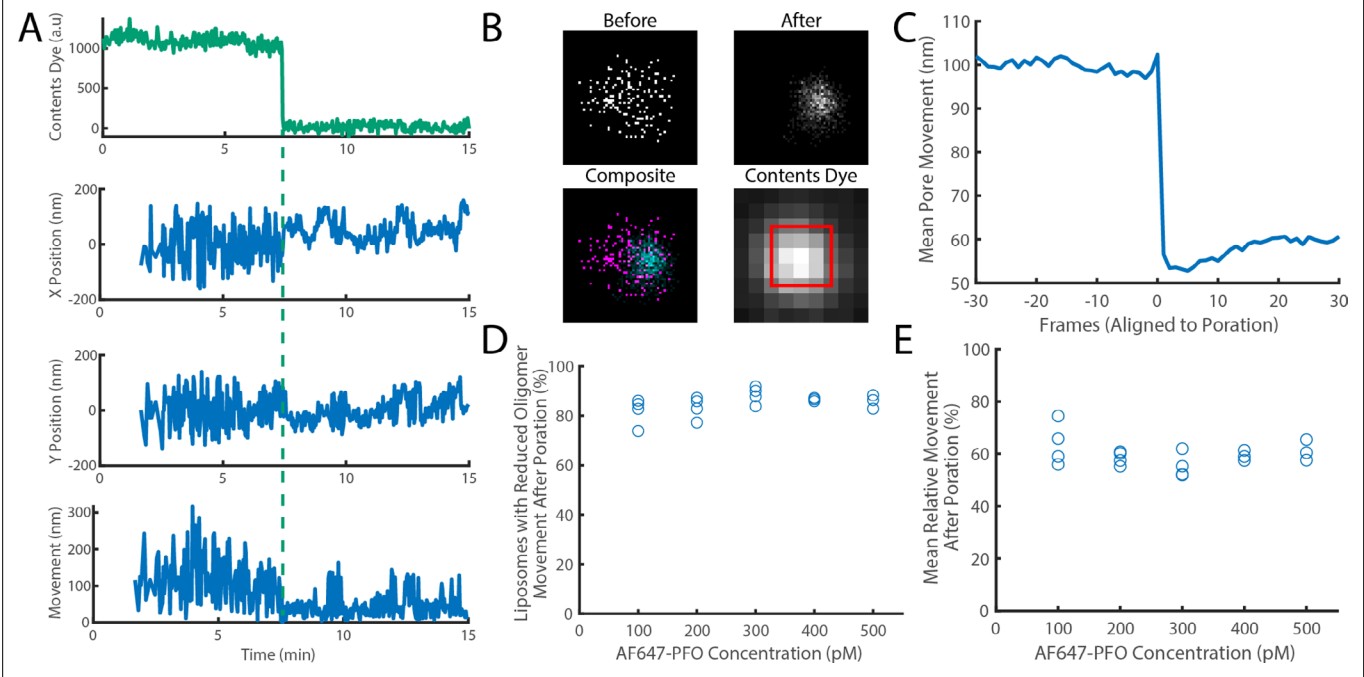

**Figure 7.** The movement of a PFO oligomer on the membrane decreases upon pore opening. Dye-loaded liposomes were imaged in the presence of 500 pM AF647-PFO by dual-colour TIRF microscopy (1 frame every 2 s, 50ms exposure time). Pore opening was detected by release of the content dye while the x/y-localisations of the PFO oligomer assembling on the liposome were tracked by point-spread function fitting. (**A**) Example traces recorded at a single liposome showing (from top to bottom) content dye intensity, x-position, y-position and movement between frames. The positions in x and y were calculated relative to the centre of the bounding box of localizations. The movement was calculated as the Cartesian distance between x/y-positions in subsequent frames. Additional examples are shown in *Figure 7—figure supplement 1*. (**B**) Maps (440 nm × 440 nm) of PFO oligomer x/y-positions acquired before (top left) and after (top right) content dye release. The overlay of both maps (bottom left) shows localisations before and after content dye release in magenta and cyan, respectively. The diffraction-limited content dye signal (bottom right) colocalises with the x/y-position maps (map area indicated by the red square). (**C**) Trace of mean PFO oligomer movement obtained from all 11905 traces in this data set aligned to the frame of dye release (poration). (**D**) The fraction of liposomes that display a reduction in their average pore movement after contents dye release as a function of concentration. (**E**) The mean change in diffusion rate at different concentrations. Four independent experiments were performed at 100, 200, and 300 pM and three experiments at 400 and 500 pM.

The online version of this article includes the following figure supplement(s) for figure 7:

**Figure supplement 1.** Example traces of PFO oligomer movement on liposomes before and after poration.

stepwise fashion upon dye release (*Figure 7C–E*). We conclude that most liposomes contain a fluorescent species that dominates the point spread functions and becomes less mobile from one frame to the next, consistent with a single oligomer becoming less diffusive upon insertion into the membrane.

## High pH inhibits PFO pore formation activity at the step of membrane binding

Above pH 7, PFO accumulation on membranes and pore formation is impaired (*Nelson et al., 2008*), but it remains unclear where in the pathway the defect occurs. Stable PFO accumulation on membranes requires oligomerisation, which depends on interactions with the membrane as well as protein-protein interactions. To dissect which interactions and whether specific steps in the PFO pore formation pathway are affected by pH, we used the dual-colour pore formation assay to measure assembly kinetics of AF647-PFO (200 pM) and dye release from liposomes between pH 5–8 in conjunction with kinetic modelling (*Figure 8*). Permeabilisation of liposomes was most efficient at pH ≤6.5 and then rapidly diminished between pH 7–8 (*Figure 8A*), whereby <3% of liposomes showed single-step dye release characteristic of pore formation at pH 8. We note that the pore formation efficiency was lower in these experiments than above (the fraction of no dye release traces is 34% in *Figure 8A* vs. 13% in *Figure 1F*), possibly due to differences between batches of AF647-PFO. In addition to

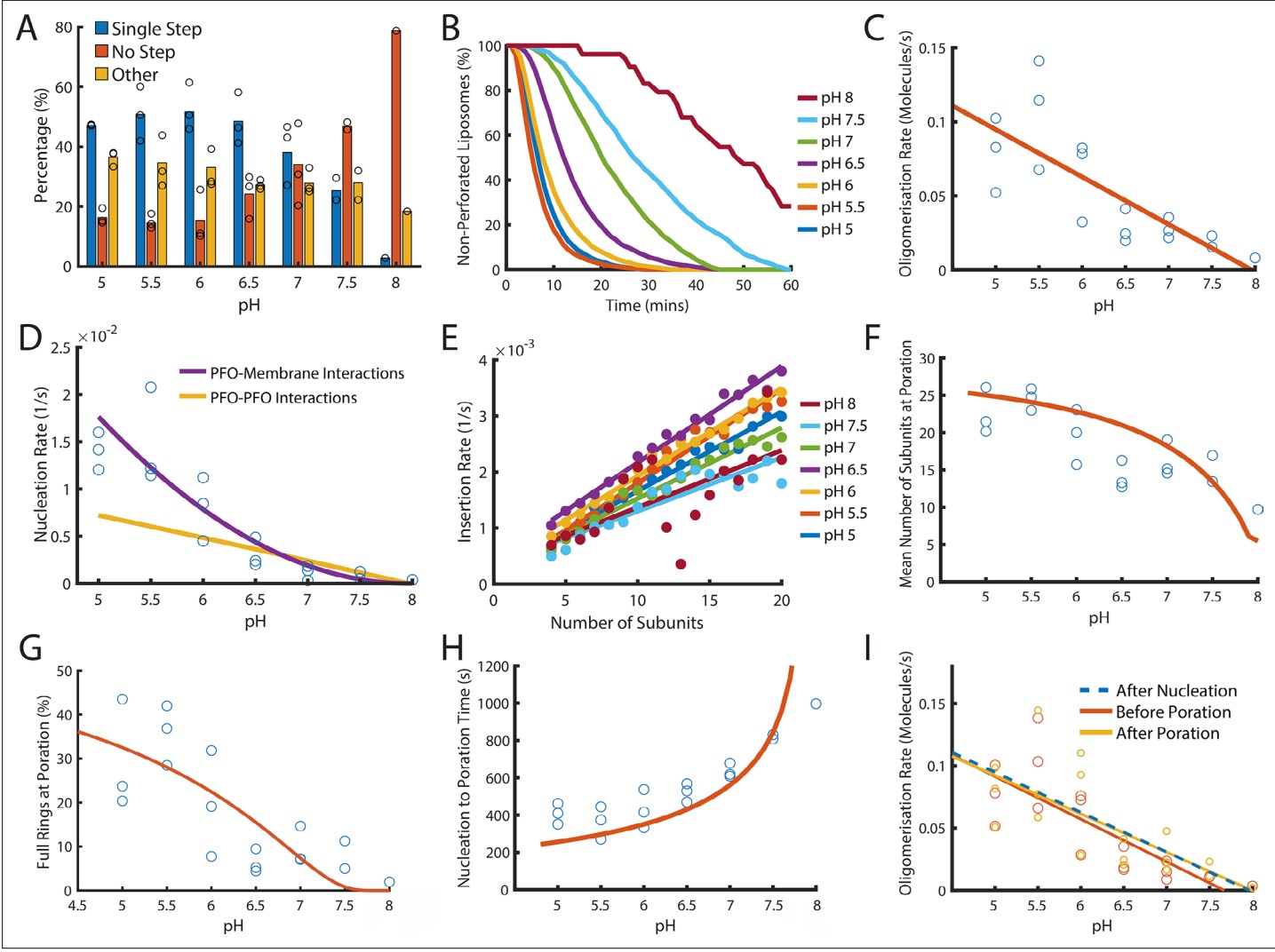

**Figure 8.** Dependence of PFO pore formation on pH. Dual-colour TIRF pore formation experiments were carried out with 200 pM AF647-PFO. (**A**) AF647-PFO pore formation activity at 200 pM decreases with increasing pH resulting in a higher proportion of liposomes with no step. Fraction of liposomes with content dye traces classified as (1) single step, (2) no/little dye loss or (3) other. (**B**) The overall rate of pore formation decreases with increasing pH. (**C**) The oligomerisation rate decreases with increasing pH. The dependence is heuristically fit with a straight line. Control experiments confirmed that the intensity of AF647-PFO was independent of pH (*Figure 8—figure supplement 2*). (**D**) The nucleation rate decreases with increasing pH. The pH dependence of oligomerisation observed in (**C**) could result from pH affecting either the membrane-binding kinetics of monomeric PFO, or the lateral interaction between membrane-bound PFO monomers. These two scenarios give rise to different predicted nucleation dependence curves shown in purple and yellow, respectively. (**E**) Changing pH does not have a significant effect on the insertion kinetics. (**F**) The mean number of subunits in the prepore at insertion decreases with increasing pH. This dependence is predicted by taking the oligomerisation rate from (**C**), suggesting that pH does not affect insertion kinetics. (**G**) The percentage of full pores at time of poration decreases with increasing pH. The dependence is predicted using the oligomerisation rate from (**B**). (**H**) The nucleation to insertion time increases with increasing pH. The dependence is predicted using the oligomerisation rate from (**B**). (**I**) Oligomerisation kinetics at different pH are the same before (red) and after (yellowed) poration. The analysis of pH dependence is based on three experiments at each pH value between pH 5–7, two experiments at pH 7.5, and one experiment at pH 8. The orange lines in panels F-H represent fits of the model in *Figure 6B* using the pH-dependent oligomerisation rates in panel C.

The online version of this article includes the following figure supplement(s) for figure 8:

**Figure supplement 1.** Nucleation kinetics decrease with increasing pH.

**Figure supplement 2.** The intensity of PFO-AF647 is independent of pH.

pH dependence of pore formation efficiency, the overall rate of pore formation slows down with increasing pH (*Figure 8B*).

Next, we inspected oligomerisation (*Figure 8C*) and nucleation (*Figure 8D*). As expected we observed a reduction in rate over the pH range, whereby nucleation was almost abolished at pH 8 (*Figure 8—figure supplement 1*). This decrease in rate may result from either reduced membrane binding of PFO monomers or from reduced lateral interactions between PFO molecules forming the oligomer. To distinguish between these two possibilities, we used the kinetic model in *Figure 4E* to predict the effect of weakening either of these interactions on nucleation (see Appendix for details). The better fit of the data by the membrane binding scenario (*Figure 8D*, purple line) leads us to conclude the observed decrease in nucleation kinetics with pH was due to a defect in PFO–membrane binding rather than a defect in PFO–PFO interactions, whereby we estimate that the affinity of PFO monomers for membranes is decreased by 10-fold between pH 5 and pH 8.

It has been proposed that the transition from the prepore to the membrane-inserted state could be affected by pH (*Rossjohn et al., 2007*). To test this possibility, we extracted the insertion kinetics as a function of the number of subunits in the oligomer as described above, which showed minor differences but no overall correlation between the insertion rate and pH (*Figure 8E*). This suggests that monomer binding was the only step affected by pH.

Next, we analysed the phase from nucleation to poration. Interestingly, the number of subunits in the arc at the time of poration was higher at pH 5 (~25 subunits) than at pH 8 (~10 subunits) (*Figure 8F*) and similarly the fraction of PFO oligomers that insert as complete rings decreased from ~30% to none detected across the pH range (*Figure 8G*). Concomitantly, the time from nucleation to poration increased with pH (*Figure 8H*). When we simply introduced the pH-dependent differences in the oligomerisation rate (*Figure 8C*) (which in turn result from the defect in membrane binding) into the oligomerisation/insertion kinetic model (*Figure 6B*), the model was able to reproduce the decrease in mean number of subunits (*Figure 8F*, orange line) and complete ring formation (*Figure 8G*, orange line) and the increase in the time from nucleation to poration (*Figure 8H*) with increasing pH (See appendix 'The effect of pH on growth and insertion' for model details). Thus, the kinetics and energetics of membrane insertion are not affected by pH. Finally, we confirmed that across the entire pH range oligomerisation continued with unchanged kinetics after arc insertion, that is the basic features of the pore assembly pathway were preserved (*Figure 8I*).

On the basis of these analyses we conclude that the overall pronounced decrease of the pore formation efficiency and kinetics with increasing pH from slightly acidic to slightly alkaline is due to reduced affinity of the monomer binding to the membrane while PFO–PFO interactions are largely unaffected. This defect propagates through other phases of the assembly pathway, slowing nucleation and oligomerisation but not the kinetics of the prepore to pore transition. As a consequence, the mean number of subunits at the time of poration decreases with increasing pH. The implication of this observation is that other parameters that affect PFO–membrane affinity (such as lipid composition) would similarly tune whether insertion occurs early during PFO arc growth or when rings are (almost) complete.

## Discussion

By measuring the single-molecule kinetics of PFO accumulation on liposomes together with a functional read-out for pore opening at the level of single liposomes, we were able to identify distinct stages in the pore formation pathway that would be undetectable in ensemble measurements due to dephasing. Our analysis provides a complete kinetic description of CDC pore assembly from monomer binding to pore opening with the following characteristics: (i) PFO monomers repeatedly bind to and diffuse on the membrane surface and dissociate with a half-life of 0.3 s; (ii) dimerisation of two PFO monomers on the membrane represents the committed step that nucleates a stable growing oligomer; (iii) oligomerisation occurs by addition of monomers that bind to the membrane from solution and is essentially irreversible; (iv) oligomers (arcs) with as few as four (possibly fewer) subunits can insert into the membrane and open a membrane pore, whereby the rate of insertion increases approximately linearly with the number of subunits in the arc; this transition from the prepore to the pore state constitutes the main energy barrier in the pathway; (v) oligomers continue to grow after insertion by addition of PFO monomers, presumably forming complete rings.

Our experimental approach relying on immobilised liposomes is complementary to AFM (**Boyd et al., 2016**; **Leung et al., 2014**; **Mulvihill et al., 2015**; **Vögele et al., 2019**) and single-molecule fluorescence microscopy (**Senior et al., 2022**) studies on flat bilayers to image the dynamic process of CDC pore assembly in real time. The assay has sufficiently high throughput to study the effect of varying experimental conditions on pore formation (in this work, PFO concentrations and pH), whereby each experiment provides traces from hundreds to over a thousand liposomes providing statistical power to detect small changes in kinetic parameters. The use of microfluidics enables temporal control of experimental conditions. In addition, the constant microfluidic solution flow facilitates experiments at exquisitely low concentrations by avoiding depletion of PFO from solution due to adsorption onto surfaces and assembly on membranes. TIRF microscopy provides sensitivity for single-molecule detection at high temporal resolution (57ms per frame) for prolonged periods of time (hours). This allowed us to measure the transient interactions of PFO monomers on liposomes at concentrations ($\leq$20 pM) where dimerisation on the membrane surface is highly improbable, providing the first estimates of the monomer binding and unbinding rates for PFO.

Limitations of our approach include the following. Unlike measurements on flat membranes, we cannot resolve separate molecules or complexes on the same liposome due to the diffraction limit. While biochemical characterisation and AFM imaging of PFPs is typically carried out in the nM range, we limited the concentrations used in this study to the pM range to enhance the probability of observing the growth of a single complex per liposome. This is initially borne out in the data: The oligomerisation rate remains constant (**Figure 4B**) instead of increasing (as would be expected for multiple oligomers) and the movement of the fluorescent complex on the membrane drops in a single step upon membrane insertion (**Figure 7C**). Nevertheless, the fluorescence signal ultimately exceeds the levels expected for a single ring-shaped pore, suggesting that additional oligomers eventually nucleate on the liposome, although it remains unclear when this second nucleation event occurs. Thus, we cannot rule out the simultaneous growth of more than one oligomer, especially during later stages of the experiment. Further experiments are required to pinpoint exactly when additional oligomers appear, for example using conditions that are optimised to detect new nucleation events or complementary approaches such as single-particle tracking on flat bilayers. Finally, it is possible that oligomer growth could also proceed by addition of dimers (or multimers) when PFO concentrations are sufficiently high for frequent dimerisation events, that is far above concentrations used in our experiments (see **Figure 6—figure supplement 5**).

Another limitation is that non-specific labelling of PFO (resulting in a heterogeneous sample of labelled species), as well as possible fluorescence artefacts, limit the precision with which we can determine the number of subunits in PFO oligomers. For validation, we determined the oligomerisation rate and number subunits in the oligomer at the time of pore opening by detecting and counting individual PFO monomers joining the growing oligomer. This intensity-independent analysis yielded essentially the same values as the quantification based on intensity.

Consistent with previous observations for PFO and other CDC/MACPF pore formers (**Leung et al., 2017**; **Leung et al., 2014**; **Senior et al., 2022**), we find that PFO oligomerisation is irreversible. Notably, our data also shows that the PFO dimer is the first long-lived species on the membrane, such that dimerisation is the committed step for pore formation. PFO does not form dimers (or oligomers) in solution (**Feil et al., 1996**; **Rossjohn et al., 1997**). On the membrane, collisions between PFO molecules may initially form weak, reversible interactions but stable PFO-PFO association is dependent on a conformational change that can only occur on the membrane surface (**Hotze et al., 2012**). This requires displacement of the β5 strand from the β-sheet in domain 3 to allow interactions between the β4 and β1′ strands of adjacent PFO molecules defining the nascent β-barrel (**Evans et al., 2020**; **Ramachandran et al., 2004**). It is worth noting that after oligomerisation, MACPF/CDC β-barrels are typically SDS-resistant and show remarkable stability (**Evans et al., 2020**; **Heuck et al., 2000**; **Shepard et al., 2000**). Our data only revealed one long-lived dimer species, suggesting that the proposed reversible dimer is too short-lived to be detected here and either dissociates or immediately converts to the stable dimer. The type of conformational change required for this transition is illustrated in **Figure 2—figure supplement 4** and **Figure 2—video 1** and resembles the pathway for dimerisation proposed on the basis of linear CDC oligomers in pneumolysin, vaginolysin, and intermedilysin crystal structures (**Lawrence et al., 2016**; **Lawrence et al., 2015**). Finally, the level of stabilisation observed here (~3 orders of magnitude increase in half-life on the membrane) is remarkable even

when accounting for avidity of a tight PFO-PFO dimer, suggesting that dimerisation goes along with further conformational changes to stabilise the membrane-bound state.

It has been shown that several MACPF/CDC PFPs can insert into the membrane as arcs before formation of the full ring is complete (*Leung et al., 2014*; *Mulvihill et al., 2015*; *Podobnik et al., 2015*; *Sonnen et al., 2014*). Our analysis showed that PFO oligomers with as few as four subunits can efficiently release the encapsulated dye from the liposome (*Figure 5*). Since PFO assembly on liposomes is not sufficient for dye release, but requires unfurling and insertion of the β-hairpins (*Heuck et al., 2003*; *Heuck et al., 2000*), our data suggests that small oligomers can insert into the membrane and induce formation of a (at least transient) semi-toroidal pore. PFO dimers and trimers do not efficiently release dye (*Figure 5*), but we cannot distinguish whether these species fail to insert into the bilayer or fail to induce membrane pore opening upon insertion. Membrane-bound monomers are short-lived, rapidly dissociating back into solution (*Figure 2*), and thus do not insert into the lipid bilayer.

Remarkably, the rate of arc insertion increased linearly with the number of subunits in the arc such that insertion of low order oligomers can only be observed in the absence of growth. One interpretation of the dependence on the number of subunits in the oligomer is that any subunit can independently trigger the process leading to insertion of the entire pore, that is the more subunits are available, the higher the probability of insertion. Overall, our interpretation fits with the sequential insertion model for β-barrel formation (*van Pee et al., 2017*), whereby the conformational change in one subunit propagates to neighbouring subunits along the oligomer. In this model, the semi-toroidal pore opens early in the process and lipids are pushed aside into the bulk membrane as insertion proceeds along the oligomer (*Vögele et al., 2019*).

Our experiments also suggested that when oligomers are continuously growing, the insertion rate increases with the concentration of PFO in solution. This unexpected observation needs further investigation and validation. While we do not yet understand the underlying mechanism, one possible hypothesis is that the binding of a monomer to the growing oligomer induces a transient activated state that is more likely to insert. More frequent binding events (observed at higher PFO concentrations in solution) would then induce the activated state more frequently, leading to an overall increase in insertion kinetics.

We observed that AF647-PFO accumulation on the liposome continued unabated after opening of an arc pore, which we interpret as post-insertion oligomer growth. Using single-molecule tracking on droplet interface bilayers supported on an agarose layer, Wallace and colleagues also observed that PFO oligomers continue to grow by monomer addition after insertion into the bilayer (*Senior et al., 2022*), whereby insertion was detected by the sudden drop in lateral diffusion of the assembling structure on the membrane. In contrast, AFM studies of suilysin pore formation on lipid bilayers supported on mica have shown that suilysin arcs do not continue to grow after insertion (*Leung et al., 2014*). Thus, further studies are needed to determine whether different CDCs and/or different experimental systems lead to different outcomes.

Real-time imaging (*Leung et al., 2017*; *Parsons et al., 2019*) and structural analysis (*Spicer et al., 2018*) suggest that some members of the MACPF family can assemble via a growing pore mechanism whereby monomers or oligomers add to an inserted arc. Notably, MACPFs do not undergo a vertical collapse upon insertion such that oligomerisation interfaces remain aligned between molecules in the prepore and the pore state. In contrast, it is unclear how membrane-bound PFO monomers in the prepore state could join a collapsed inserted arc and how this interaction could lead to insertion of the newly arrived PFO molecule since insertion is thought to depend on the formation of extensive PFO-PFO contacts (*Burns et al., 2019*; *Wade et al., 2015a*). While further experiments are required to establish whether post-insertion PFO oligomerisation adds monomers to the already inserted arc pore to complete the ring or whether the arc pore facilitates nucleation of a separate oligomer, it is nevertheless tempting to speculate how post-insertion growth may occur. One hypothesis is that the PFO monomer may interact with the inserted arc, for example through initial D4–D4 interactions (*Harris et al., 1991*), followed by vertical collapse and unfurling of the β-hairpin (*Figure 6—figure supplement 3*). This model is akin to the sequential unfurling of β-hairpins discussed above and requires considerable conformational flexibility (possibly of the subunits at the edge of the arc). An alternative hypothesis is that incoming PFO monomers may undergo an induced conformational change, for example as a result of interacting with the toroidal lipid edge, and subsequently add to the arc pore (*Figure 6—figure supplement 3*). All of our models for post-insertion oligomerisation require PFO to

undergo conformational changes, driven through local environmental effects. While this is precisely what drives conventional PFO oligomerisation, characterising the specific details in the context of inserted arcs remains an unresolved question, whereby each hypothesis presented here has puzzling elements but provides the basis for future experimental studies.

As discussed above, insertion can be observed over a wide range of oligomer sizes, ranging from ~tetramers to full rings. We propose a model in which the transition from arc or ring-shaped prepore to pore is a stochastic process that is controlled by the kinetics of the different steps in the pore formation pathway. For example, under conditions where membrane binding is slow (e.g. at low PFO concentrations or pH >7), oligomerisation is too slow to complete a full ring before insertion occurs. Since PFO monomers can continue to add after insertion, the final outcome is a ring-shaped pore, regardless of when insertion happened during arc growth. This is consistent with a predominance of ring pores in EM images, even when PFO assembly occurs at low concentrations (*Figure 6—figure supplement 4*). While this model is sufficient to explain the data, additional mechanisms may be operational such as the proposed allosteric trigger for the conformational change upon ring closure (*Wade et al., 2015a*). In addition to arcs and rings, pores can also be formed when two (or more) arcs coalesce into larger structures (*Leung et al., 2014*; *Mulvihill et al., 2015*; *Ruan et al., 2016*; *Senior et al., 2022*). We expect the relative abundance of these structures to be controlled by kinetics as well. For example, if PFO monomers are depleted from solution during the reaction, arc growth slows down such that coalescence becomes more likely.

Overall, our single-molecule approach enables analysis of all steps leading to PFO pore formation to be deconvoluted and investigated separately, unlike existing ensemble assays. The assay design is simple to modify and enables multiple experimental modalities for different lines of inquiry. Further, the moderate throughput enables statistical power to quantify individual processes within the full assembly pathway. In conjunction with kinetic modelling, this analysis represents a useful method to predict and measure the effects of modification to specific steps in the pathway, with applications in drug development or biotechnology (*Johnstone et al., 2021*).

## Methods

### PFO production and purification

PFO C459A without signal peptide containing an amino-terminal hexahistidine tag (sequence shown in the Appendix; referred to as PFO herein) was used in this work. PFO C459A lacks the cysteine residue prone to oxidation (leading to inactivation) and has the same pore-forming activity as wild type PFO (*Pinkney et al., 1989*; *Saunders et al., 1989*; *Shepard et al., 1998*). The protein was expressed in *E. coli* BL21DE3 cells at 37 °C using a codon-optimised sequence cloned into the pET-15b plasmid. Cells were lysed in 20 mM HEPES pH 7.5, 150 mM NaCl, 20 mM imidazole, 0.1% Triton-X 100, protease inhibitor (Sigma) and DNAse I (Sigma). Cell debris was separated from the soluble fraction by centrifugation at 10,000 $xg$ for 30 min at room temperature. The protein was purified by $Ni^{2+}$ IMAC chromatography on a HisTrap column (Cytiva). The column was washed using 20 mM HEPES pH 7.5, 150 mM NaCl, 20 mM imidazole and bound protein eluted using 20 mM HEPES pH 7.5, 150 mM NaCl, 500 mM imidazole. Eluted protein was dialysed overnight into 25 mM HEPES pH 7.5, 150 mM NaCl, 2.5% glycerol and further purified by size exclusion chromatography on a SEC Superdex S200 16/60 column (Cytiva). Peak fractions were concentrated and stored at –80 °C.

### PFO labelling

PFO was dialysed twice for 1 hour against HBS pH 7.5 (20 mM HEPES, 100 mM NaCl) using a Slide-A-Lyzer Mini dialysis capsule (10 k MWCO) and the protein concentration (determined using the absorbance at 280 nM) was adjusted to 24 µM. The labelling reaction was carried out in the dark at room temperature. An aqueous solution of Alexa Fluor 647 (AF647) NHS ester (2.5 mM) was added to a concentration of 120 µM. The long-wavelength AF647 dye was chosen for PFO as it is less prone to self-quenching than short-wavelength dyes. After a reaction time of 1 h, HisPur Ni-NTA resin (40 µL) was added to the mixture. Beads were collected by centrifugation and washed with HBS pH 8 (3×50 µL). After elution with a 1:1 (v:v) mixture of 1 M imidazole (pH 7) and HBS (pH 7) for 10 min, the beads were pelleted by centrifugation and the supernatant containing AF647-PFO was collected. The protein concentration was determined using the Bradford assay. AF647-PFO was analysed by

SDS-PAGE with fluorescence densitometry and the degree of labelling was determined to be ~1.5 fluorophores/PFO by comparison with a labelled protein standard. Analysis of AF647-PFO adsorbed onto a coverslip showed that the majority of PFO molecules (~60–85% depending on batch) were singly labelled, while the remainder had more than one label (*Figure 2E*). The membrane poration activity of AF647-PFO was essentially the same as that of unlabelled PFO (*Figure 1—figure supplement 2*).

## Lipid film preparation and liposome extrusion

Solutions of cholesterol, 1-palmitoyl-2-oleoyl-glycero-3-phosphocholine (16:0-18:1 POPC) and 1-(12-biotinyl(aminododecanoyl))–2-oleoyl-sn-glycero-3-phosphoethanolamine (12:0 Biotin-18:1 PE) in chloroform (10 mg/mL, Avanti Polar Lipids) were mixed in a 55:44:1 molar ratio. The solution was divided into 120 µL aliquots in glass vials, dried down under vacuum, overlaid with nitrogen and stored at –40 °C until use.

The lipid film was allowed to equilibrate to room temperature, rehydrated in 250 µL HBS pH 7 containing 0.01 mg/mL AF488 and resuspended by forcefully passing the mixture several times through a 25 G needle. The sample was vortexed and subjected to 6 freeze-thaw cycles using liquid nitrogen. Liposomes were extruded using a LiposoFast liposome factory by passing the solution an uneven number of times (21–31) through a 0.22 µM polycarbonate membrane filter. The extruded liposomes were passed through a Sephadex G-25 column to remove free dye.

## Fabrication and operation of microfluidic flow cells

Microfluidic fabrication and TIRF assay adapted from *Márquez et al., 2018*. Glass coverslips were cleaned by sonication in ethanol for 30 min followed by sonication in 1 M NaOH for 30 min, rinsed with ultrapure water and dried. PDMS devices for assembly of microfluidic flow cells (channel height 60 µm, channel width 800 µm) were prepared using standard protocols for soft lithography. After treating the PDMS device and the coverslip with an air plasma inside a plasma cleaner for 5 min, the PDMS device was mounted on the coverslip and the assembled microfluidic flow cell was heated in an oven at 70 °C for at least 15 min. A second treatment with air plasma was carried out to improve bonding between the glass and the PDMS. The glass surface at the bottom of the microfluidic channels was then modified by adsorption of a co-polymer composed of poly-L-lysine (PLL) and biotinylated poly(ethylene glycol) (PEG) (Susos AG, PLL(20)-g[3.4]-PEG(2)/PEG(3.4)-biotin (20%)). A solution of PLL-PEG-biotin (0.1 mg mL⁻¹ in PBS) was injected into the flow channels and incubated at room temperature for 5 min followed by flushing the channels with water and drying. The channels were then filled with a solution of streptavidin (Sigma-Aldrich, 0.2 mg mL⁻¹) for 15 min and rinsed with HBS pH 7. The channels were rinsed with isopropanol to remove air bubbles prior to treatment with blocking buffer (20 mM Tris pH 7.5, 2 mM EDTA, 50 mM NaCl, 0.03% NaN3, 0.025% Tween 20, 0.2 mg mL⁻¹ BSA). The microfluidic flow cell was mounted on the microscope stage and connected to tubing. Solutions were pulled through the channels using a syringe pump connected to the outlet tubing and operating in 'withdraw' mode.

## TIRF microscopy assays

AF488-loaded liposomes in HBS (30–100 µL) were flowed through the microfluidic channel and captured on the modified coverslip via interaction of the biotinylated lipids with the surface-bound streptavidin. Unbound liposomes were washed out with 50 µL of HBS pH 7. All experiments were carried out at room temperature unless stated otherwise.

Images were collected on a custom built TIRF microscope based around an ASI-RAMM frame (Applied Scientific Instrumentation) with a Nikon 60×CFI Apochromat TIRF (1.49 NA) oil immersion objective. Solid State Lasers were incorporated using the NicoLase system (*Nicovich et al., 2017*). Images were captured on three Photometrics Prime BSI Cameras (Teledyne Photometrics). A total of 250-mm tube lenses were used to give a field of view of 176 µm × 176 µm. Alternatively, single molecule binding images were collected on a TIRF microscope based around a Nikon Eclipse Ti-E2 chassis equipped with a Nikon 100×CFI Apochromat TIRF (1.49 NA) oil immersion objective, solid state lasers for excitation, and a single Photometrics Prime 95B Camera.

## Single molecule experiments

Experiments to measure AF647-PFO monomer binding were acquired as follows. HBS pH 7 (5 mL) containing AF647-PFO (2.5–15 pM) and BSA (0.01 mg/mL) as a blocking agent was constantly flowing

through the channel at a rate of 10 μL/min. A total of 432,000 single molecule binding TIRF images were acquired for each independent experiment using a 20ms exposure and 100 mW 647 nm laser power at a frame rate of 57ms per frame.

Experiments to measure the long-lived species were acquired as follows. HBS pH 7 (5 mL) containing AF647-PFO (10 pM) and BSA (0.01 mg/mL) was constantly flowing through the channel at a rate of 7 μL/min. A total of 2000 images of AF647-PFO were acquired using a 200ms exposure, 3 mW 647 nm laser power at a frame rate of 20 s per frame. In between AF647-PFO images, images of the liposome content dye were captured using 20ms exposure and 3 mW 488 nm laser power. The liposome images were used to correct for stage drift during acquisition as AF647-PFO binding was too sparse to reliably calculate drift in these experiments.

Single molecule binding was detected using the Localize program in the Picasso suite (*Schnitzbauer et al., 2017*) with a minimum net gradient set to 250 for detection of monomers and to 150 for detection of long-lives species. The box size was set to 9 pixels. Localisations were filtered for particles with a localisation precision of less than 1 in both x and y directions to exclude cases where the background was erroneously detected. Single molecule binding experiments were drift corrected using the redundancy cross-correlation function built into the Render program of the Picasso suite with an averaging window size of 10,000 frames. Individual binding events were linked using a maximum link localisation distance of 7 and a maximum number of dark frames of 5.

## Liposome diameter measurements using super-resolution TIRF microscopy

Super-resolution images were exported as a grayscale image from the Render Picasso program with a fixed 5-fold up-sampling (giving an effective pixel size of 22 nm). Particles were detected by thresholding and liposome sizes were calculated using a Feret diameter.

## PFO concentration titration

HBS pH 7 containing AF647-PFO (50–500 pM), AF488 (100 nM) as a solution exchange marker and BSA (0.01 mg/mL) as a blocking agent was constantly flowing through the channel at a rate of 20 μL/min. TIRF images were acquired (488 nm laser and 640 laser, 50ms exposure time). The total duration of the experiment varied depending on the pore-formation kinetics, lasting between 15 and 240 min, whereby the total number of frames was kept at 180 frames.

## Image analysis

Image stacks of PFO assembly/dye release experiments were analysed using home-written image analysis software (JIM v4.3, freely available at https://github.com/lilbutsa/JIM-Immobilized-Microscopy-Suite; copy archived at swh:1:rev:39a402c8dddcad16b15b7b5aa5994b407ec6fe3f [*Walsh, 2021*]). Channel alignment and drift correction were performed as a part of the JIM trace generation pipeline using a C++implementation of a subpixel cross-correlation algorithm (*Guizar-Sicairos et al., 2008*). Traces exhibiting loss of the AF488 signal in one step were included in the analysis while multi-step traces, traces without dye release or otherwise uninterpretable traces were excluded from analysis. The time of PFO addition (or PFO wash-out) was detected as an overall increase (or decrease) of the background fluorescence marker. The number of bound AF647-PFO molecules was determined from the ratio of the AF647-PFO fluorescence intensity associated with the liposome to the fluorescence intensity of a single AF647-PFO molecule. The fluorescence intensity of the single fluorophore was determined from the quantal photobleaching step in photobleaching traces of AF647-PFO molecules adsorbed sparsely to the coverslip surface and imaged continuously.

## TIRF imaging of oligomerisation by detecting arrival of single AF647-PFO molecules

TIRF imaging for detection of the arrival of single AF647-PFO molecules by step fitting (*Figure 4—figure supplement 3*) and for tracking the localisation of AF647-PFO oligomers by point spread function fitting (*Figure 7*) were conducted with increased 640 nm laser power of 75 mW and an exposure time of 50ms exposure. Frame rates were as follows (frames per second, FPS): 100 pM, 1/9 FPS; 200 pM, 1/4.5 FPS; 300 pM, 1/3 FPS; 400 pM, 1/2.5 FPS; 500 pM, 1/2 FPS. Step fitting of the AF647-PFO traces was performed using change point analysis in Matlab with a threshold of $3\times10^5$. The

x/y-positions of AF647-PFO oligomers diffusing on liposomes were calculated using Picasso Localize (box size 9, minimum net gradient 800). Image stacks in both channels were corrected for stage drift using the cross-correlation calculated from the dye contents channel in JIM. The clusters of x/y--localisations of AF647-PFO oligomers detected using Picasso were aligned with the regions of interest of liposomes in the dye channel in JIM to generate dual-colour traces.

## PFO wash-out experiment

HBS (pH 7 and 0.01 mg/mL BSA) containing AF647-PFO (400 pM) and AF488 (100 nM) as a solution exchange marker was constantly flowing through the channel at a rate of 20 µL/min for 5 min. Flow was stopped and inlet tubing was swapped to a HBS (pH 7 and 0.01 mg/mL BSA) solution without PFO or dye and flow was resumed at 20 µL/min less than a minute after stoppage. Dual-colour TIRF imaging was conducted using a 488 nm laser and a 640 laser with 50ms exposure time. To avoid photobleaching in experiments used to measure the number of subunits in oligomers, imaging only commenced following the washout of free PFO in solution. The total duration of the experiment varied depending on the pore-formation kinetics, lasting between 15 and 240 min, whereby the total number of frames was kept at 180 frames. The temperature was controlled using an OkoLab Bold Line stage top incubator.

## Kinetic models

The construction of the kinetic models is described in detail in the Appendix.

## Acknowledgements

We thank Quill Bowden, Shaghayegh Baghapour and Chantal Márquez for help and discussion during early stages of assay development. We thank Rodney Tweten for helpful discussions. This work was supported by NHMRC Project Grant APP1182212 (TB), ARC Future Fellowship FT150100049 (MD), NHMRC Investigator Grant APP1194263 (MWP), Australian Research Council Grants DP160101874 and DP200102871 (MWP and CJM) and a UNSW International Postgraduate Award (CM). Infrastructure support from the NHMRC Independent Research Institutes Infrastructure Support Scheme and the Victorian State Government Operational Infrastructure Support Program to St. Vincent's Institute are gratefully acknowledged. MWP is an NHMRC Leadership Fellow.

## Additional information

### Funding

| Funder | Grant reference number | Author |
| --- | --- | --- |
| National Health and Medical Research Council | APP1182212 | Till Böcking |
| Australian Research Council | FT150100049 | Michelle A Dunstone |
| National Health and Medical Research Council | APP1194263 | Michael W Parker |
| Australian Research Council | DP160101874 | Michael W Parker |
| Australian Research Council | DP200102871 | Craig J Morton |

The funders had no role in study design, data collection and interpretation, or the decision to submit the work for publication.

### Author contributions

Conall McGuinness, Formal analysis, Investigation, Methodology, Writing – original draft; James C Walsh, Conceptualization, Software, Formal analysis, Supervision, Investigation, Methodology, Writing – original draft; Charles Bayly-Jones, Formal analysis, Investigation, Writing – review and

editing; Michelle A Dunstone, Supervision, Funding acquisition, Writing – review and editing; Michelle P Christie, Resources, Writing – review and editing; Craig J Morton, Writing – review and editing; Michael W Parker, Resources, Funding acquisition, Writing – review and editing; Till Böcking, Conceptualization, Supervision, Funding acquisition, Methodology, Writing – original draft

**Author ORCIDs**
James C Walsh http://orcid.org/0000-0003-0447-2323
Charles Bayly-Jones http://orcid.org/0000-0002-7573-7715
Craig J Morton http://orcid.org/0000-0001-5452-5193
Michael W Parker http://orcid.org/0000-0002-3101-1138
Till Böcking http://orcid.org/0000-0003-1165-3122

**Decision letter and Author response**
Decision letter https://doi.org/10.7554/eLife.74901.sa1
Author response https://doi.org/10.7554/eLife.74901.sa2

---

## Additional files

### Supplementary files
• Transparent reporting form

### Data availability

The image analysis software is available at https://github.com/lilbutsa/JIM-Immobilized-Microscopy-Suite (copy archived at swh:1:rev:39a402c8dddcad16b15b7b5aa5994b407ec6fe3f). Microscopy image stacks for Figure 1 and Figures 3-8; files containing single-molecule tracks extracted from all image stacks for Figure 2 (single-molecule binding); and a representative subset of image stacks recorded for Figure 2 are available on Dryad (doi: https://doi.org/10.5061/dryad.8w9ghx3q4). The complete set of image stacks collected for Figure 2 is too large (>10 TB) to be included in this repository such that these data are stored on the UNSW data archive (data management plan number D0240569) and can be obtained for research (including commercial) by submitting a request to research.soms@unsw.edu.au.

The following dataset was generated:

| Author(s) | Year | Dataset title | Dataset URL | Database and Identifier |
| --- | --- | --- | --- | --- |
| Walsh J, Boecking T | 2022 | Single-molecule analysis of the entire perfringolysin O pore formation pathway | https://doi.org/ 10.5061/dryad. 8w9ghx3q4 | Dryad Digital Repository, 10.5061/dryad.8w9ghx3q4 |

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

## Appendix 1

### PFO amino acid sequence

The amino acid sequence of the His-tagged PFO C459A protein used in this study is shown below. The N-terminal hexahistidine tag followed by a thrombin cleavage site (underlined) are highlighted in blue. This construct for recombinant PFO expression in *E. coli* used in this study is missing the N-terminal signal peptide sequence (1-28) of the PFO precursor that is cleaved off during the biosynthesis of the functional protein in *Clostridium perfringens*. The numbering of the PFO residues is according to the full-length precursor protein (i.e. the construct for recombinant expression starts at K29).

    MGSSHHHHHH SSGLVPRGSH MKDITDKNQS IDSGISSLSY NRNEVLASNG DKIESFVPKE 60
    GKKAGNKFIV VERQKRSLTT SPVDISIIDS VNDRTYPGAL QLADKAFVEN RPTILMVKRK 120
    PININIDLPG LKGENSIKVD DPTYGKVSGA IDELVSKWNE KYSSTHTLPA RTQYSESMVY 180
    SKSQISSALN VNAKVLENSL GVDFNAVANN EKKVMILAYK QIFYTVSADL PKNPSDLFDD 240
    SVTFNDLKQK GVSNEAPPLM VSNVAYGRTI YVKLETTSSS KDVQAAFKAL IKNTDIKNSQ 300
    QYKDIYENSS FTAVVLGGDA QEHNKVVTKD FDEIRKVIKD NATFSTKNPA YPISYTSVFL 360
    KDNSVAAVHN KTDYIETTST EYSKGKINLD HSGAYVAQFE VAWDEVSYDK EGNEVLTHKT 420
    WDGNYQDKTA HYSTVIPLEA NARNIRIKAR EATGLAWEWW RDVISEYDVP LTNNINVSIW 480
    GTTLYPGSSI TYN 493

### Mathematical analysis of PFO pore formation

#### Nucleation analysis

In *Figure 3F*, the dimer is shown to only be metastable, with a half-life of 13 min, although the pathway of dimer dissociation is not identified. The two potential methods of dimer dissociation, the dimer falling directly off the membrane in one step, or a two-step process, where the dimer falls apart into two membrane monomers which can then individually dissociate are shown in *Appendix 1—figure 1* in the middle and right-hand schematics respectively. Here we calculate the nucleation rates for these two scenarios and compare them to the scenario the dimer was irreversibly bound (*Appendix 1—figure 1A* Left).

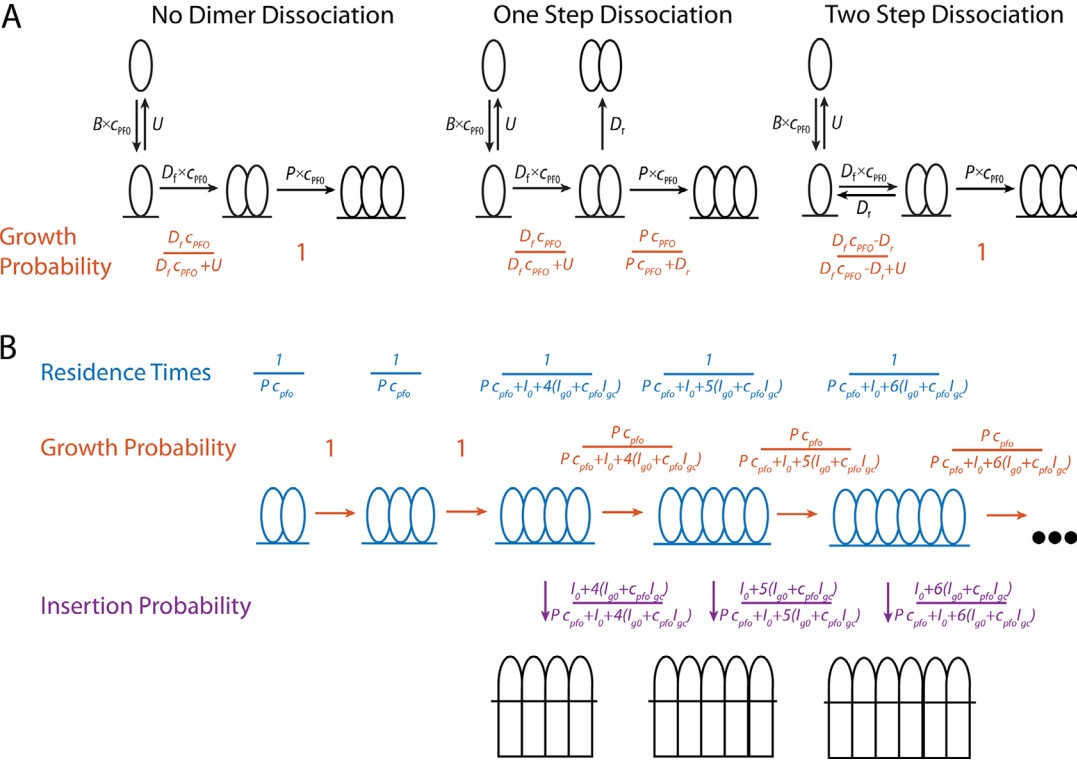

**Appendix 1—figure 1.** Mathematical modelling of PFO. (**A**) Schematics of the nucleation pathway for different scenarios of dimer dissociation (**B**) The residence times, and transition probabilities, of each state in the pore formation pathway. These are used to calculate the distribution of the number of subunits in the prepore at the time of insertion as well as the nucleation to insertion times.

The probability that a monomer will become a dimer and a dimer will become a trimer is shown in orange below each transition in *Appendix 1—figure 1A*. In general, the probability that a single molecule will grow rather than fall off the membrane is given by the rate of growth divided by the sum of all rates out of that state. The one exception to this is in the two-step model where the rate of dimerisation is substituted by an effective rate of dimerisation ($D_f\, C_{pfo} - D_r$).

The nucleation rate can be modelled as the rate of monomer binding to the membrane ($B\, c_{pfo}$) multiplied by the probability that the monomer forms a dimer then forms a trimer. For no dimer dissociation this gives a rate of:

$$\text{Nucleation rate} \;=\; B\, c_{pfo} \frac{D_f\, c_{pfo}}{D_f\, c_{pfo} + U}$$

For a single step dimer dissociation the nucleation rate is

$$\text{Nucleation rate} \;=\; B\, c_{pfo} \frac{D_f\, c_{pfo}}{D_f\, c_{pfo} + U} \frac{P\, c_{pfo}}{P\, c_{pfo} + D_r}$$

For a two-step dimer dissociation the nucleation rate is

$$\text{Nucleation rate} \;=\; B\, c_{pfo} \frac{D_f\, c_{pfo} - D_r}{D_f\, c_{pfo} - D_r + U}$$

These equations are plotted in *Figure 4—figure supplement 2B*. The difference between the two release scenarios and the no dissociation scenario are shown in *Figure 4—figure supplement 2C*. For values above 100 pM the difference is less than 5%, suggesting that the dimer off rate is not a significant factor during nucleation.

## Insertion analysis
Following nucleation PFO oligomerises into an arc (or potentially a full ring) before inserting into a ring. At each oligomer size (in molecules), the pore formation process can be thought of as a

choice, either the oligomer adds an additional monomer and grows, or it inserts to form a pore. The likelihood of each of these choices depends on how quickly the oligomer is growing in number of subunits, and how fast it is able to insert. The higher the solution concentration of PFO, the faster it will oligomerise, and the more likely it is to grow rather than insert. Conversely, the more subunits in the prepore, the faster it inserts (*Figure 5C* and *Figure 6A*).

Mathematically, the probability of whether a prepore oligomer will grow, or insert, is given by the rate of oligomerisation ($P\,c_{pfo}$), or insertion ($I_0 + n\,(I_{g0} + c_{pfo}\,I_{gc})$) respectively, divided by the sum of these two rates. This is shown schematically in *Appendix 1—figure 1B* with growth probabilities coloured in Orange and insertion probabilities shown in Purple.

Overall, the probability that a pore will insert with a given number of subunits is then given by the product of the probabilities that the arc grew for all sizes less than the number of subunits at insertion before inserting at that number of subunits. For example, the probability of a pore inserting as a 5mer is:

$$Prob\,(5) \;=\; 1\;\times 1\;\times\;\frac{P\,c_{pfo}}{P\,c_{pfo}+I_0+4\left(I_{g0}+\,c_{pfo}\,I_{gc}\right)}\;\times\;\frac{I_0+5\left(I_{g0}+\,c_{pfo}\,I_{gc}\right)}{P\,c_{pfo}+I_0+5\left(I_{g0}+\,c_{pfo}\,I_{gc}\right)}$$

This can be generalised for any number of subunits, n as:

$$Prob\,(n) \;=\; \frac{I_0+n\left(I_{g0}+\,c_{pfo}\,I_{gc}\right)}{P\,c_{pfo}+I_0+n\left(I_{g0}+\,c_{pfo}\,I_{gc}\right)}\;\prod_{j=m}^{n-1}\;\frac{P\,c_{pfo}}{P\,c_{pfo}+I_0+j\left(I_{g0}+\,c_{pfo}\,I_{gc}\right)}$$

$$= I_0^{m-n-1}\left(I_0 + n\left(I_{g0} +\, c_{pfo}\,I_{gc}\right)\right)\left(P\,c_{pfo}\right)^{n-m}/pch\left(\left(I_0 + m\,\left(I_{g0} +\, c_{pfo}\,I_{gc}\right) + P\,c_{pfo}\right)/\left(I_{g0} +\, c_{pfo}\,I_{gc}\right),n\right)$$

Where *m* is the minimum number of subunits for insertion (assumed to be 4) and *pch* is the pochhammer function.

The size distribution predicted by this equation for various concentrations is shown in *Figure 6—figure supplement 2B*. These distributions are directly comparable to the experimental distributions shown in *Figure 6—figure supplement 2A*.

Any oligomers that reach 35 subunits are assumed to form complete rings and cannot oligomerise any further, only insert. The percentage of oligomers that form a complete ring is shown in *Figure 6D*. The mean of the number of subunits at insertion distribution is plotted as a function of concentration in *Figure 6C* where it is also compared to the mean of experimental data.

## Nucleation to insertion time

During the oligomerisation process, we can calculate the average amount of time that an oligomer will spend with a given number of subunits before deciding to either grow or insert. The average total time from nucleation to insertion for a pore with a given number of subunits can then be calculated by summing these times.

The mean residence times for each state is given by the inverse of the sum of rates out of that state. This is shown schematically in blue in *Appendix 1—figure 1B*. The average time take for an oligomer to reach a certain number of subunits is then given by the sum of the times it spends in each size up to that number of subunits:

$$Growth\;Time\,(n) \;=\; \frac{m-2}{Pc_{pfo}} + \sum_{j=m}^{n}\frac{1}{Pc_{pfo}+I_0+j\left(I_{g0}+\,c_{pfo}\,I_{gc}\right)}$$

To generate a mean nucleation to insertion time for all pores, the mean insertion time for each number of subunits is weighted by the probability of insertion occurring with that number of subunits (calculated in the Insertion Analysis section). A plot of the mean nucleation to insertion time for all pores as a function of concentration is shown in *Figure 6E*.

## The effect of pH on nucleation

The pH dependence of the PFO oligomerisation rate shown in *Figure 8C* was heuristically fit with a straight line and taken as:

$$P\left(\left[pH\right]\right) = 0.243 - 0.03\left[pH\right]$$

The pH dependence could result from either the membrane-binding kinetics of monomeric PFO, or the lateral interaction between membrane-bound PFO monomers being affected by changes in pH.

In the scenario where pH affects membrane binding (or release) then both the membrane binding and polymerisation interactions are affected. The value of single molecule binding ($B$) was scaled to change proportionately to the change in polymerisation:

$$B\left([pH]\right) \ = \ \frac{B\left([pH]=7\right)}{D_f\left([pH]=7\right)}D_f\left([pH]\right) = 0.95 - 0.12\left[pH\right]$$

Note that since single molecule binding to liposomes is transient ($U >> D\,c_{pfo}$) either single molecule binding ($B$) or unbinding ($U$) can be scaled and give the same results.

These two rates were then substituted into the nucleation rate equation from the section Nucleation Analysis:

$$Nucleation\ rate\left([pH]\right) \ = \ B\left([pH]\right)\ c_{pfo}\frac{P\left([pH]\right)c_{pfo}}{P\left([pH]\right)c_{pfo}+U}$$

This line is plotted in purple in **Figure 8D**.

In the case that pH only affects PFO-PFO lateral interactions, then only the polymerisation rate would be affected so the equation would then be:

$$Nucleation\ rate\left([pH]\right) \ = \ B\ c_{pfo}\frac{P\left([pH]\right)c_{pfo}}{P\left([pH]\right)c_{pfo}+U}$$

This line is plotted in yellow in **Figure 8D**.

## The effect of pH on growth and insertion

The dependence of insertion kinetics was modelled by substituting the heuristic oligomerisation rate dependence fit from **Figure 8C** (See "The Effect of pH on Nucleation"):

$$P\left([pH]\right) = 0.243 - 0.03\left[pH\right]$$

Into the equations for insertion distribution (see 'Insertion Analysis') and nucleation to insertion time (see 'Nucleation to Insertion Time'). The concentrations were then set to equal experiments (200 pM) and rates were plot as a function of pH to generate curves shown in **Figure 8F–H**.

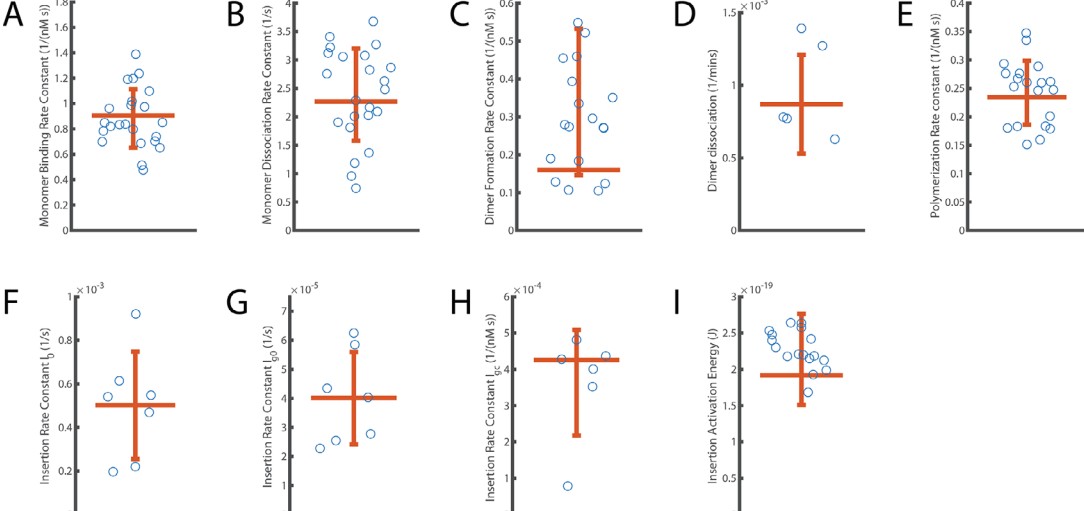

**Appendix 1—figure 2.** Distribution of independent measurements for each parameter. The error bar displays the standard deviation of measurements around the mean. The horizontal orange line shows the aggregated fit value reported for each parameter. (**A**) Monomer binding rate constant ($B$; from **Figure 2**). (**B**) Monomer unbinding rate ($U$; from to **Figure 2**). (**C**) Dimer formation rate constant ($D_i$; from **Figure 3**). (**D**) Dimer dissociation rate ($D_r$; from **Figure 3**). (**E**) Oligomerisation rate constant ($P$; from **Figure 4**). (**F–H**) Components describing the overall kinetics of insertion during continuous oligomer growth: insertion rate ($I_0$, shown in panel F), oligomer subunit number-dependent insertion rate ($I_{g0}$, shown in panel G) and PFO concentration-dependent insertion rate ($I_{gc}$, shown in

*Appendix 1—figure 2 continued*

panel H) (from *Figures 5 and 6*). (**I**) Activation energy for insertion, that is transition from the prepore state to the open pore state ($E_a$; from *Figure 5*).

## Error calculation of parameter estimates

### Single-molecule photobleaching for intensity calibration

A single-molecule photobleaching experiment was used to calculate conversion factors to convert measured fluorescent intensities to numbers of bound molecules. Measuring the photobleaching rate also governed how many frames were acquired before bleaching becomes significant.

The protocol for photobleaching was as follows:

1. 25 mm round coverslips were cleaned by sonication in ethanol, water, 1 M NaOH then water again for 15 min each before being blow-dried with nitrogen.
2. A coverslip was exposed to an air plasma using a plasma cleaner (Harrick Plasma) before being placed in a Chamlide chamber to prevent liquid from running off the edge of the slide.
3. 1 mL of the fluorescently labelled PFO at 50 pM was added to the coverslip and left to bind for 5 min.
4. The supernatant was removed and the sample was washed with 1 mL clean buffer to remove unbound molecules.
5. Wash buffer was then discarded and replaced with fresh wash buffer.
6. The sample was then imaged on the microscope (*Appendix 1—figure 3*). A photobleaching image stack was collected by exposing a field of view with the same laser power setting used during the actual experiment but at four times the exposure time (200ms).
7. 200 frames were imaged so that approximately 90% of fluorophores were bleached.
8. Five fields of view were imaged to measure variability within the sample and consistency of analysis.

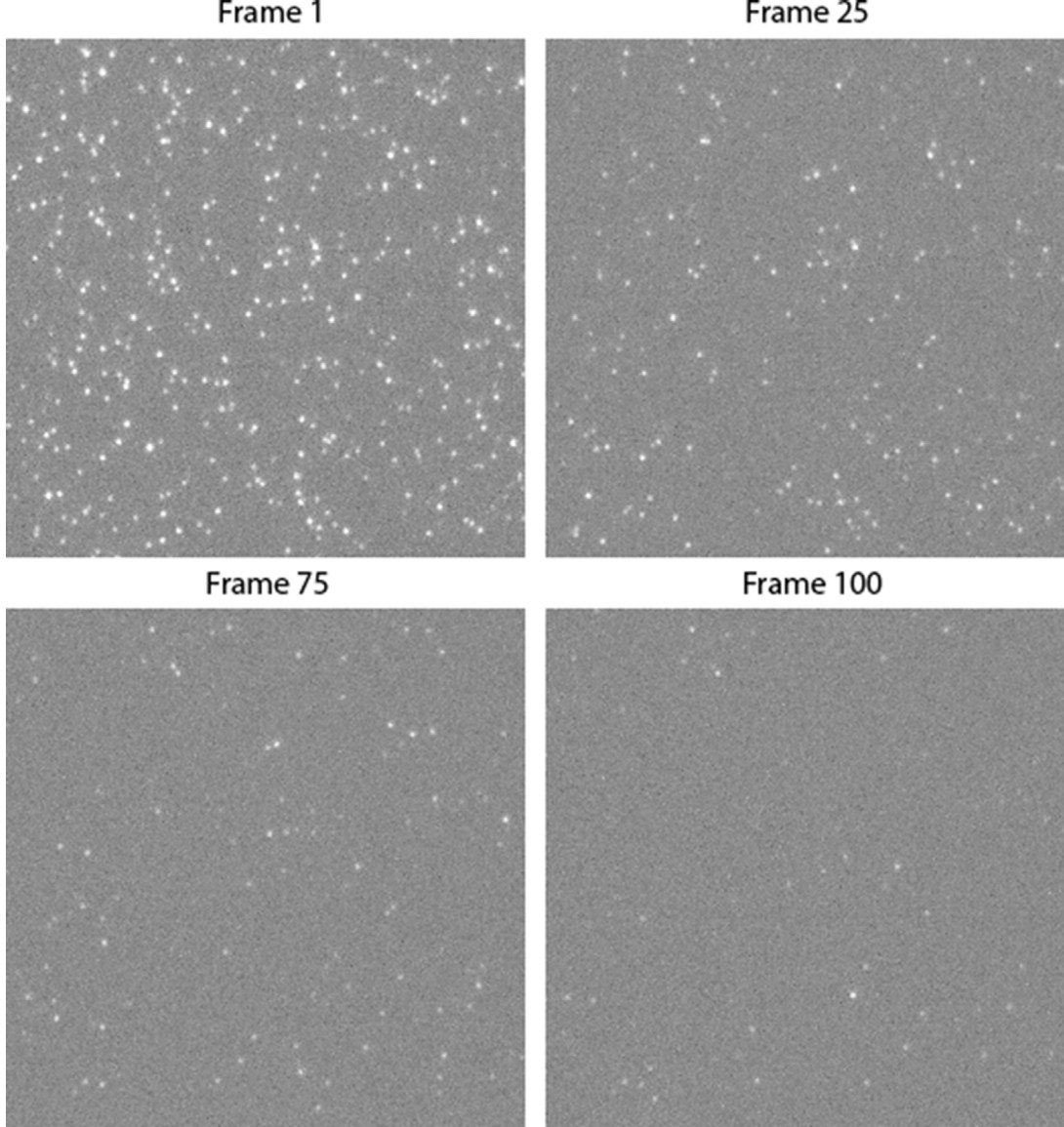

**Appendix 1—figure 3.** Montage of single-molecule photobleaching image stack. Single PFO particles, immobilised on a coverslip, bleach as more frames are imaged. The montage shows the 512x512 pixel centre of a single field of view. The complete image is 2048x2048 and in total 5 fields of view were imaged.

## Photobleaching image analysis

Image stacks are analysed using the JIM software package developed in our lab. This software is open source, continuously maintained, and freely available at https://github.com/lilbutsa/JIM-Immobilized-Microscopy-Suite; *Walsh, 2021*. The documentation for JIM contains multiple tutorials including a step by step guide for collecting and analysing single-molecule photobleaching. The documentation is available at https://docs.google.com/document/d/12frP6jp74eiycXxY8knR_qb27ui7Ia-RNMBuAuAHRFs/edit

Traces are generated for photobleaching image stacks by detecting regions of interest (ROIs) from the first 10 frames of the experiment. These traces are then analysed using the step fitting program (part of the JIM package) which uses change point analysis to heuristically determine whether or not a step occurs in a trace. An in-depth description of the step fitting program is available in the relevant section of the documentation.

After step fitting, three filters are applied to the step traces to determine which traces to include in the analysis. These filters are as follows:

- first step probability: 0.5
- ratio between the second and first intensity level: 0.25 (at least 75% of the signal is lost)
- probability of more steps: 0.999 (exclude traces with additional steps)

The photobleaching rate is obtained by fitting an exponential decay function to the distribution of bleaching times (*Figure 2—figure supplement 3A, B*). The single-molecule intensity is obtained from the mean of a log-normal curve fitted to the distribution of step heights (*Figure 2D*).

