## [Editor Report]

This paper presents a detailed single-molecule, multi-color microscopy study of the real-time assembly of perfringolysin O, a member of the membrane attack complex perforin cholesterol-dependent cytolysin superfamily. With the ability to resolve different reaction species simultaneously with membrane leakage, this work provides key mechanistic details including identifying assemblies involved in membrane lysis, and how membrane binding, oligomerization, and pore transitioning depends on concentration and pH. This study will be of interest to many, particularly those studying cytolysin mechanisms, but also the broader field of single-molecule studies of membrane binding proteins.

---

## [Decision Letter]

**Decision letter after peer review:**

Thank you for submitting your article "Single-molecule analysis of the entire perfringolysin O pore formation pathway" for consideration by *eLife*. Your article has been reviewed by 3 peer reviewers, including Janice L Robertson as Reviewing Editor and Reviewer #1, and the evaluation has been overseen by Nancy Carrasco as the Senior Editor. The following individual involved in review of your submission has agreed to reveal their identity: Ana J Garcia-Saez (Reviewer #3).

Essential revisions:

The reviewers found this to be an intriguing study, offering exquisite resolution into the complex reaction of perfringolysin O pore formation on membranes. However, all of the reviewers raised concerns about controls that are needed to interpret the data. It is anticipated that the following essential revisions can be reasonably addressed with further elaboration of the methods, revised analysis of the current data, or a limited number of control experiments. In general, the manuscript should also be revised to consider alternate interpretations of the data.

1) Mapping corrections between the co-localization fields. For co-localization of single-molecules, an experimental mapping correction is often performed in order to calculate the translational and magnification transformations that may be needed to map a pixel from one emission field to the other. This can be calculated using a sample that is known to contain co-localized molecules and imaged in both fields. Were experimental mapping corrections carried out in this study? In the following, it sounds that this mapping is carried out by the software: "We then used a software developed for single-molecule localisation microscopy (Schnitzbauer et al., 2017) to detect and track AF647-PFO spots appearing and disappearing at the locations of individual liposomes." If this is the case, please elaborate on how this works and whether you have data confirming that this method provides an adequate correction.

2) Estimating the number of PFO molecules from intensities. Since the single-molecule intensities follow a broad distribution (Figure 2D), it is likely that any estimation of the number of PFOs from intensities will have a large uncertainty and will have to include consideration of different labeling probabilities. Along these lines, the PFO quantities should be reported as the raw AF647 intensities in the figures. A clear description of how the number estimations are calculated along with the uncertainty on these estimates can be included in the discussion, as these number values are rough estimations.

3) The effect of photobleaching. A main result of this work is that the dimer has a much higher association stability with the membrane than monomers. However, monomers and dimers were measured with widely different power intensities and at different frame rates. Hence, it should be absolutely clear to the reader that this observation does not simply reflect the different bleaching rates of the two different molecules. For example, the monomer was sampled with high wattage powers and at 17 frames per second (˜1000 per minute). The dimers are sampled at high wattage (how high?) and 3 frames per minute. Considering that the measured dwell time for monomers is 0.5 s, while the one for dimers is ˜20 min (or 1200 s), it follows that if the authors are measuring the bleaching of the molecule rather than the association of the molecules with the membranes, then there would actually be no difference between the association of monomers and dimers.

To address this, more evidence is required to show that what is being measured is association and not bleaching. Specifically, what is the bleaching rate of the dye under the experimental conditions? This could be easily measured for the dimer, as single-step intensity reductions of the fluorescence should provide information on bleaching. The authors might also consider using different dyes with different bleaching rates and compare the association rates. In addition, they say that the signal disappeared in one step, which is indicative of dissociation of the dimer from the membrane, rather than a photobleaching step (perhaps the authors should mention this). It is assumed that occasionally, the signal disappeared in two steps. Please indicate how many times a two-step process was observed. The kinetics of the two-step process can also be used to characterise the photobleaching time. Please indicate here how fast is photobleaching and how that compares with the dwell time of the monomer binding. Comparison of rates measured at different fluorescent power of frames per seconds might provide a quick way to establish what is measured here.

4) Possibility of multiple complexes on a single liposome. One weakness of the study is the inability to distinguish between one or more complexes simultaneously assembling in one liposome. It is unclear how one can distinguish whether the estimated PFO numbers reflect the stoichiometry of one oligomer and not multiple oligomers on one liposome? The continued growth would suggest that this cannot be distinguished. Are there any abrupt steps in the oligomerization process that could be attributed to additional pore structures growing from defined seeding points? The single-step dye loss was interpreted to mean that only one pore is likely to permeate the liposome (Pg. 6), with multi-step signals interpreted as being from multilamellar liposomes. While plausible, there is no clear evidence of this, and it remains possible that the multi-step increases are due to more than one pore forming in the liposome. To clarify this, it would be beneficial to measure the full width at half maximum of the AF647-PFO fluorescence intensity profiles on individual liposomes and plot them as a normal distribution. This would allow to exclude measurements with a full width at half maximum outside the population average (e.g. the 95th percentile of the distribution) which most likely correspond to liposomes containing more than on PFO complex. In addition, the paper requires revisions to consider the alternate possibility that the data reflect multiple pore forming reactions on the single liposomes.

5) Please address the recommendations raised by the reviewers below. These suggestions are directed at clarifying the manuscript and interpretation of the results.

*Reviewer #1 (Recommendations for the authors):*

1. In Figure 1D, it will be useful to add examples of the "single step", "no step", and "other" traces directly in this main figure, as has been depicted in Figure 1 —figure supplement 3. Visualization of these representative traces is important to allow the reader to properly interpret the different behaviors.

2. In 2G, the super-resolution approach for PFO monomer binding appears to outline the liposome size quite well. Can you quantify the liposome size distribution based on this approach? Do you find that there is a significant amount of non-uniformity to indicate that your results may depend on liposome size?

3. Addition of uncertainty and statistical information. Much of the data presented lacks a description of the uncertainty or variability. For example, in the frequency/percentage bar plots, the binomial standard deviation could be used. In other plots, the mean is shown but without standard deviation. Please add this information and provide details in the legends about population or sample numbers.

4. Revise the following paragraph on page 4 as this paper is no longer accompanying:

"In an accompanying paper, Wallace and colleagues use single-molecule fluorescence tracking of PFO assembly on a droplet interface bilayer (Senior et al., 2021). These imaging studies support the insertion of incomplete arc-shaped membrane lesions, suggesting an alternative mechanism of pore formation that is distinct from the canonical prepore formation prior to insertion. This discrepancy raises the question as to when and how release of the membrane spanning regions is triggered, which cannot be correlated with key assembly steps using ensemble methods. Related to this matter, it is also unclear whether release and insertion of the membrane spanning regions from each of the subunits occurs in a concerted or sequential fashion."

5. Further technical details would be useful in this paper, such as how the co-localization was determined and corrected for different mappings between fields, as well as how the number of PFO molecules was ascertained by intensity analysis.

*Reviewer #2 (Recommendations for the authors):*

1. Page 7. How was the surface area of the liposome established?

2. Page 9-10. The authors go to a great length to describe the kinetics of oligomer formation. However, it is not clear how the authors can actually distinguish between the formation of many independent dimers and the formation of oligomers. Do they assume that once the dimers are on the membrane they will eventually meet up into oligomers? Is this a reasonable assumption given the diffusion time of the dimers and the size of the liposome?

3. Furthermore, it is not clear how the authors can distinguish between the formation of conductive arc pores and full pores. Since it is only possible to observe a full liposome, all these species should have the same fluorescence properties. How can the authors know the number of subunits of a pore, considering that the fluorescent signal can only estimate the overall number of units in the membrane? How can the authors distinguish a full pore from two arcs?

4. Page 10. The authors argue that the arc pores with at least four subunits can form pores. How did they come to this conclusion? Only on the basis of fluorescence intensity? If that is the case, has photobleaching taken into account? The authors identified the oligomer-length specific insertion rates from the insertion times. However, it is not clear how this was done. The authors should either point to the methods where more details should be given, or explain more in detail how this was done (e.g. is the number of subunit estimated by the fluorescence intensity?).

5. The kinetic model (Figure 6B) also assumes that a monomer is added to a nucleated pore. However, since monomers are transient and dimers rather stable, would it make more sense that dimers are constantly formed from monomers and these structures move around the membrane until they meet each other and then form full pores? Would it also not be easier for a full pore to be formed from two arc pores? The authors should spend a few arguments to convince the reader that the experimental setting can distinguish between these different scenarios?

6. Pore insertion kinetics (Page 12). This connects to the arguments above. How can the authors distinguish between the growing of arc pores from the formation of unrelated structures in the same liposome? Why this cannot be explained with a new nucleation process.

7. The authors observed a marked pH dependence of pore formation. How is the fluorescent signal affected by the pH? Please also indicate where the reader can find the mathematical models that allow distinguishing between membrane and lateral interactions.

8. Furthermore, the authors should measure directly (on and off rates) the monomer and dimer affinity for the liposomes at different pH values by fluorescence.

9. Figures. Please add the condition (salt, buffer and pH) in the legend of the figures.

*Reviewer #3 (Recommendations for the authors):*

Overall, the quality of the experiments and their analysis is excellent, and I have only some questions to be addressed:

1. The authors attribute the heterogeneous release profiles to heterogeneity in the liposome preparation. What about the stochastic nature of pore formation? That would also support the distributions observed between time of nucleation/poration.

2. The fluorescence signal of PFO on individual liposomes seems heterogeneous, which the authors attribute to differences in cholesterol distribution. However, the lipid composition is not expected to present phase separation. Could this be due to complex assembly on part of the liposome leading to heterogeneous distribution? Otherwise, the authors might test their hypothesis of inhomogeneous cholesterol distribution by fluorescence microscopy on giant unilamellar vesicles or lipid bilayers containing a lipophilic dye.

3. Please explain how the different kinetics of monomer addition to the pre-pore or the pore states are considered in the mathematical model.

4. Please explain how the oligomerization rate by monomer addition before and after insertion can be the same. One would expect that the energetics of the interactions of the monomers with the non-inserted and inserted oligomers would differ. This part of the model is not very convincing yet and requires elaboration.

5. Do the complete rings continue to grow indefinitely? Please explain.

6. While the authors nicely confirm that AF647-labelled PFO is functional and that the effect of self-quenching is small using different ratios of labelled to unlabeled PFO (Figure 1 —figure supplement 2), this analysis also revealed that a 100% ratio of labelled PFO is more active in perforating liposomes. While this effect can be caused by an underestimation of the amount of unlabeled protein (as the authors mention), they should also discuss the option that AF647-PFO is indeed more active due to possible structural alterations caused by the label.

7. The authors use photobleaching step counting of sparsely distributed AF647-PFO immobilized on glass coverslips to measure the reference monomer fluorescence intensity (as shown in Figure 1E and D). Considering that AF647-PFO has an estimated labeling efficiency of ~1.5 fluorophores/PFO, it would be beneficial to compare the population distribution of the determined monomer intensities (Figure 1E) to the intensity distribution measured for monomeric AF647 dye immobilized on glass.

8. In order to measure the oligomeric state of AF647-PFO at different time points in the liposomes, the authors determine the ratio of the fluorescence intensity associated with the liposome to the monomer fluorescence intensity derived from single photobleaching events. Given that not all PFO molecules are labelled with one single AF647 fluorophore (Figure 2E), the calculation of molecular units from fluorescence intensity is somewhat error prone. Labelling efficiency correction of the measured fluorescence intensities should be used to test if there is an influence of the labelling efficiency on the determined oligomeric state of AF647-PFO.

9. Considering the lipid composition of the outer leaflet of the plasma membrane (as a biological target for PFO pore formation), a concentration of 55% cholesterol seems very high. The authors should state in the manuscript why they chose the specific lipid mixture and how it might affect the mechanistic regulation of PFO (membrane binding, oligomerization, etc.) compared to the physiological scenario of the plasma membrane.

10. When discussing the possible explanations why PFO pores continue to grow over time with wit a membrane insertion rate increasing with concentration, the authors should also discuss the effect of continuous protein insertion on the physical properties of a toroidal pore. Increasing number of protein molecules inserted in a membrane increase the membrane tension and at the same time decrease the line tension at the pore rim both promoting the stabilization of an open pore.

11. In this line, the authors should also include a figure showing the oligomerization characteristics of PFO (as in Figure 4 B) beyond the expected number of molecules required to form a full pore ring until the oligomerization reaches saturation. The possible explanation for the continuous increase in the number of PFO molecules that "eventually multiple pores form" (page 6) is contradictory to the valid assumption made later that an existing PFO oligomer in the liposome "acts as a sink for monomers" (page 12) that newly bind to the membrane.

12. The authors should state in the manuscript why a Cysteine-free version of PFO (PFO(C459A)) was used.

13. Quantify the percentage of unpermeabilized liposomes in Figure 2 —figure supplement 2.

14. Please provide statistics (i.e. number of measured vesicles or complexes, number of technical replicates and test for statistical significance if applicable) for the data shown in Figure 1 D and E, Fig, 2 E and D, and Figure 1 —figure supplement 1.

15. It is misleading that PFO oligomers with a number of subunits below the reported number needed to form a full ring are referred to as arcs. There is no experimental prove in this study that these oligomeric states of PFO indeed assemble to an arc-like structure in the membrane.

16. The terminology "oligomer length" implies a structural parameter. Better refer to it as "molecular size" / "oligomeric state2 / "no of subunits/molecules".

17. Please clarify that Figure 3D shows the sum of individual localizations over time and not multiple simultaneous localizations of AF647-PFO to one liposome.

18. The deliberate offset of the two fluorescence channels in Figure 2 —figure supplement 2 is somewhat misleading as it hinders the correct interpretation of the data. It would be better to show the correctly aligned image of the overlay and the single channel images and the merge of the cropped area for better visualization.

19. Please specify the meaning of the orange line in Figure 7 F and H.

20. Page 8: Please specify in the sentence "To determine the stoichiometry of AF647-PFO in the long-lived state, we measured the average intensity of all long-lived signals on liposomes." that it is averaging over time and not over several particles.

21. Please include scale bars in the microscopy images in Figure 1C, Figure 2C, Figure 2B, Figure 2 —figure supplement 1, and Figure 2 —figure supplement 2.

22. Refer to Table 1 also in the main text of the manuscript.

23. Page 13: Please correct the referencing to the different panels in Figure 7 F-I.

24. Page 6: Please correct the typos "Figur1 Figure Supplement 3" and "Figur1 Figure Supplement 3B".

25. Figure 3D: Please correct the typo "AF4647-PFO" in Figure 3D.

---

## [Author Response]

Essential revisions:1) Mapping corrections between the co-localization fields. For co-localization of single-molecules, an experimental mapping correction is often performed in order to calculate the translational and magnification transformations that may be needed to map a pixel from one emission field to the other. This can be calculated using a sample that is known to contain co-localized molecules and imaged in both fields. Were experimental mapping corrections carried out in this study? In the following, it sounds that this mapping is carried out by the software: "We then used a software developed for single-molecule localisation microscopy (Schnitzbauer et al., 2017) to detect and track AF647-PFO spots appearing and disappearing at the locations of individual liposomes." If this is the case, please elaborate on how this works and whether you have data confirming that this method provides an adequate correction.

A. PFO assembly and dye release experiments in Figures 1, 4, 5 and 6:

Channel alignment and drift correction were performed as a part of the JIM trace generation pipeline using a C++ implementation of a subpixel cross-correlation algorithm (Manuel Guizar-Sicairos, Samuel T. Thurman, and James R. Fienup, "Efficient subpixel image registration algorithms," Opt. Lett. 33, 156-158 (2008)). This implementation uses the Intel Math Kernel Library (MKL) and Intel Performance Primitives (IPP) for optimization but is otherwise completely open-source and freely available as a part of the Github repository. Details on usage of the image alignment code are found in the “Basic C++ Header Libraries“ section of the JIM documentation (see link below). Validation of the channel alignment and drift correction calculations is provided in Tutorial 1 and Tutorial 2 of the JIM Analysis Suite respectively using artificially generated datasets. In general, drift correction is accurate within 0.1-0.5 pixels which is sufficient for most applications.

Link to JIM documentation:

https://docs.google.com/document/d/12frP6jp74eiycXxY8knR_qb27ui7Ia-RNMBuAuAHRFs/edit?usp=sharing

B. Single molecule binding experiments in Figures 2 and 3:

AF647-PFO image stacks were drift corrected using the redundancy cross-correlation function built into the Render program of the Picasso suite with an averaging window size of 10,000 frames. The reconstructed super-resolution image of AF647-PFO localisation was manually aligned with the diffraction-limited liposome image (drift corrected using JIM as above) for visualisation (e.g. Figure 2—figure supplement 1). We note that the liposome image was not used for quantification in these experiments.

C. PFO polymer movement experiments in Figure 7:

Image stacks were corrected for drift using JIM as described above. The two channels were then aligned using JIM as described above.

The methods section of the manuscript has been updated accordingly.

2) Estimating the number of PFO molecules from intensities. Since the single-molecule intensities follow a broad distribution (Figure 2D), it is likely that any estimation of the number of PFOs from intensities will have a large uncertainty and will have to include consideration of different labeling probabilities. Along these lines, the PFO quantities should be reported as the raw AF647 intensities in the figures. A clear description of how the number estimations are calculated along with the uncertainty on these estimates can be included in the discussion, as these number values are rough estimations.

Reporting of intensities:

As suggested, the raw AF647-PFO intensity (a.u.) is now shown in graphs of single-liposome PFO binding traces. The downstream analysis uses the average of thousands of individual traces such that the high variability of individual fluorophores is averaged out and the quantification is robust and reproducible.

How was the number of PFO molecules estimated from intensity?

A description detailing the process of estimating protein numbers has been added to the Appendix under the heading “Single-Molecule photobleaching for intensity calibration”.

New experiments to validate the PFO number estimates:

To further validate the use of a single molecule intensity value, we performed additional experiments to independently measure the oligomerization rate without using the single-molecule intensity value. To do this, we imaged PFO binding to liposomes at a fast frame rate and with high laser intensity to observe individual molecules binding to the oligomer. The AF647-PFO intensity trace was then fitted with a step function to identify incoming PFO molecules. These experiments provided intensity-independent estimates of the oligomerization rate, the number of PFO monomers in the oligomer at the time of membrane insertion (detected via dye release) and the fraction of oligomers that reach the requisite number of subunits to insert as closed rings. The values for these parameters obtained by intensity calibration in Figure 4C and Figure 6C/D are within a factor of 1.5 or better to the values obtained by step fitting. We have described the new analysis based on step fitting to detect the arrival of single molecules in the Results section (page 9 and page 12), the Methods section and added the data as Supplementary Figure 4—figure supplement 3.

Results (pg 9, pg 12)

PFO dimerisation on the membrane nucleates a stably growing oligomer “Finally, we validated the oligomerisation rate by directly observing the arrival of AF647-PFO monomers using single molecule imaging conditions, which yielded a value of 0.15±0.04 nM-1 s^-1^ (Figure 4—figure supplement 3). Importantly, this measurement is independent of the single-molecule intensity calibration.”

Pore insertion kinetics for continuously growing arcs “TIRF imaging at higher laser power and temporal resolution to detect the addition of single PFO monomers to oligomers yielded similar values to the ones described above for the number of PFO monomers in the oligomer at the time of membrane insertion and the fraction of oligomers that reach the requisite number of subunits to insert as closed rings (Supplementary Figure 4—figure supplement 3E and F). This analysis provides an intensity-independent validation of these parameters.”

Methods (pg 23) "TIRF imaging for detection of the arrival of single AF647-PFO molecules by step fitting (Figure 4–Figure Supplement 3) and for tracking the localisation of AF647-PFO oligomers by point spread function fitting (Figure 7) were conducted with increased 640 nm laser power of 75 mW and an exposure time of 50 ms exposure. Frame rates were as follows (frames per second, FPS): 100 pM, 1/9 FPS; 200 pM, 1/4.5 FPS; 300 pM, 1/3 FPS; 400 pM, 1/2.5 FPS; 500 pM, 1/2 FPS. Step fitting of the AF647-PFO traces was performed using change point analysis in Matlab with a threshold of 3x105. The x/y-positions of AF647-PFO oligomers diffusing on liposomes were calculated using Picasso Localize (box size 9, minimum net gradient 800). Image stacks in both channels were corrected for stage drift using the cross-correlation calculated from the dye contents channel in JIM. The clusters of x/y-localisations of AF647-PFO oligomers detected using Picasso were aligned with the regions of interest of liposomes in the dye channel in JIM to generate dual-colour traces.”

3) The effect of photobleaching. A main result of this work is that the dimer has a much higher association stability with the membrane than monomers. However, monomers and dimers were measured with widely different power intensities and at different frame rates. Hence, it should be absolutely clear to the reader that this observation does not simply reflect the different bleaching rates of the two different molecules. For example, the monomer was sampled with high wattage powers and at 17 frames per second (˜1000 per minute). The dimers are sampled at high wattage (how high?) and 3 frames per minute. Considering that the measured dwell time for monomers is 0.5 s, while the one for dimers is ˜20 min (or 1200 s), it follows that if the authors are measuring the bleaching of the molecule rather than the association of the molecules with the membranes, then there would actually be no difference between the association of monomers and dimers.To address this, more evidence is required to show that what is being measured is association and not bleaching. Specifically, what is the bleaching rate of the dye under the experimental conditions? This could be easily measured for the dimer, as single-step intensity reductions of the fluorescence should provide information on bleaching. The authors might also consider using different dyes with different bleaching rates and compare the association rates. In addition, they say that the signal disappeared in one step, which is indicative of dissociation of the dimer from the membrane, rather than a photobleaching step (perhaps the authors should mention this). It is assumed that occasionally, the signal disappeared in two steps. Please indicate how many times a two-step process was observed. The kinetics of the two-step process can also be used to characterise the photobleaching time. Please indicate here how fast is photobleaching and how that compares with the dwell time of the monomer binding. Comparison of rates measured at different fluorescent power of frames per seconds might provide a quick way to establish what is measured here.

What is the laser power used for imaging dimers on membranes?

The nominal laser power is 100 mW for monomers and 3 mW for dimers. These values are stated in the Methods section.

What is the bleaching rate in the experiments and how does it compare to the rate of monomer release?

We have performed the additional control experiments suggested by the reviewers to show that monomer unbinding is not the result of photobleaching and that we have correctly compensated for photobleaching. This analysis has been added as Figure 2—figure supplement 3.

1. Single-molecule photobleaching experiments at a range of laser powers (Figure 2—figure supplement3A-C) show that the bleaching rate (panel B) and the single-molecule intensity (panel C) is proportional to the laser power.

2. We performed single-molecule AF647-PFO binding experiments at a range of laser powers (Figure2—figure supplement 3D/E) and corrected the rate of signal disappearance at each laser by the corresponding photobleaching rate determined above. This analysis shows that the monomer unbinding rate is independent of laser power in the range between 25–100 mW (Figure 2—figure supplement 3F), We conclude that the monomer unbinding rate is not affected by photobleaching.

3. For the dimer, we had already shown that changing the frame rate by a factor of 6 does not change the dissociation rate (Figure 3–Supplement 1) strongly suggesting that the release of the dimer from the membrane is not the result of photobleaching.

We have referred to the new Supplementary Figure showing that the monomer unbinding rate is independent of photobleaching in the Results section.

Results (pg 7)

Characterisation of PFO monomer binding to liposomes “An exponential fit of this distribution then provided the AF647-PFO monomer unbinding rate after correcting for photobleaching (Figure 2—figure supplement 3).*”*

4) Possibility of multiple complexes on a single liposome. One weakness of the study is the inability to distinguish between one or more complexes simultaneously assembling in one liposome. It is unclear how one can distinguish whether the estimated PFO numbers reflect the stoichiometry of one oligomer and not multiple oligomers on one liposome? The continued growth would suggest that this cannot be distinguished. Are there any abrupt steps in the oligomerization process that could be attributed to additional pore structures growing from defined seeding points? The single-step dye loss was interpreted to mean that only one pore is likely to permeate the liposome (Pg. 6), with multi-step signals interpreted as being from multilamellar liposomes. While plausible, there is no clear evidence of this, and it remains possible that the multi-step increases are due to more than one pore forming in the liposome. To clarify this, it would be beneficial to measure the full width at half maximum of the AF647-PFO fluorescence intensity profiles on individual liposomes and plot them as a normal distribution. This would allow to exclude measurements with a full width at half maximum outside the population average (e.g. the 95th percentile of the distribution) which most likely correspond to liposomes containing more than on PFO complex. In addition, the paper requires revisions to consider the alternate possibility that the data reflect multiple pore forming reactions on the single liposomes.

Are there features in the AF647-PFO traces that could be attributed to nucleation of additional pores?

The image acquisition settings for these experiments were optimised for measuring the kinetics between nucleation and pore opening and we prefer not to interpret trends in the data beyond this range with respect to the timing of new nucleation events. Instead, reliable detection of new nucleation events would require different settings and/or complementary approaches such as imaging on flat bilayers. Please see response to point #11 from reviewer #3 for more details.

Could multi-step traces arise from the opening of multiple pores on the same liposome?

No. We have simulated dye release via diffusion through pores using FEM modelling, showing that the dye is completely released on a time scale (0.1–1 ms) well below the temporal resolution of our measurements (Figure 1—figure supplement 4). Thus, opening of the first membrane is expected to lead to release of the entire dye contents in a single frame. Please see response to point #1 from reviewer #3 for more details.

Consider the possibility that the data reflect multiple pore forming reactions on the single liposomes.

The experimental approach using liposomes is particularly suited to measure the early stages of the first pore nucleating on the liposome and we agree that it is difficult to distinguish the nucleation of additional independent oligomers that may occur at later stages of the process. We have added new experiments and theoretical considerations to address this question (see below). We have also modified the Discussion to better highlight the limitations of our study with respect to identifying additional nucleations.

Discussion (pg 16/17)

“Limitations of our approach include the following. Unlike measurements on flat membranes, we cannot resolve separate molecules or complexes on the same liposome due to the diffraction limit. While biochemical characterisation and AFM imaging of PFPs is typically carried out in the nM range, we limited the concentrations used in this study to the pM range to enhance the probability of observing the growth of a single complex per liposome. This is initially borne out in the data: The oligomerisation rate remains constant (Figure 4B) instead of increasing (as would be expected for multiple oligomers) and the movement of the fluorescent complex on the membrane drops in a single step upon membrane insertion (Figure 7C). Nevertheless, the fluorescence signal ultimately exceeds the levels expected for a single ring-shaped pore, suggesting that additional oligomer eventually nucleate in the liposome, whereby it remains unclear when this second nucleation event occurs. Thus, we cannot rule out the simultaneous growth of more than one oligomer, especially during later stages of the experiment. Further experiments are required to pinpoint exactly when additional oligomer appear, e.g. using conditions that are optimised to detect new nucleation events or complementary approaches such as single-particle tracking on flat bilayers.

Another limitation is that non-specific labelling of PFO (resulting in a heterogeneous sample of labelled species), as well as possible fluorescence artefacts, limit the precision with which we can determine the number of subunits in PFO oligomers. For validation, we determined the oligomerisation rate and number of subunits in the oligomer at the time of pore opening by detecting and counting individual PFO monomers joining the growing oligomer. This intensity-independent analysis yielded essentially the same values as the quantification based on intensity.”

New experiments to test for the simultaneous growth of two (or more) independent PFO oligomers.

To further test whether multiple PFO oligomers are present on the liposome at the time of insertion, we have performed additional PFO binding/dye release TIRF experiments in which we track the movement of the AF647-PFO signal between frames. These experiments show a sudden decrease of movement that coincides with dye release. This observation is consistent with the insertion (and concomitant decrease in diffusion) of a PFO oligomer that dominates the fluorescence signal. These experiments are shown in a new Figure 7 and are described in the Results section.

Results (pg 13)

Pore opening coincides with a drop in PFO oligomer movement on the membrane

“Membrane insertion leads to a drastic reduction of the diffusion rate of CDC complexes on the surface of flat membranes (Senior et al., 2021); (Leung et al., 2014). To determine whether the same effect could be observed on liposomes, we imaged PFO assembly and pore formation assay with high laser intensity to accurately localise the position (projected onto the x/y-plane) of growing PFO oligomers on liposomes over time with sub-pixel resolution. The example trace in Figure 7A shows that the frame-to-frame movements of the AF647-PFO signal initially fluctuated between 0–300 nm (Figure 7A). Pore opening (detected by dye release) led to a pronounced drop in movement (Figure 7A after 7.5 minutes). This reduction in movement is also evident in the map of all x/y-localisations. Localisations appeared as a diffuse point cloud with approximately the same size as the liposome (Figure 7B top left versus bottom right), and formed a tight focus upon dye release.

Experiments conducted at 100–500 pM AF647-PFO showed that oligomers on 85% of liposomes showed a reduction of ~41% in movement across the entire concentration range, which occurred in a stepwise fashion upon dye release (Figure 7C-E). We conclude that most liposomes contain a fluorescent species that dominates the point spread functions and becomes less mobile from one frame to the next, consistent with a single oligomer becoming less diffusive upon insertion into the membrane.”

Theoretical considerations to estimate the probability for nucleation of a second PFO oligomer.

A key question for interpreting the continuing increase of the AF647-PFO intensity after insertion is whether a second PFO oligomer is likely to nucleate on the liposome when an actively growing oligomer is already present on the membrane. To address this question, we have calculated the probability of a second dimer forming on the membrane before insertion of the first oligomer (new Figure 6—figure supplement 5C). This analysis shows that in the lower end of the concentration range, the majority of liposomes contain only a single actively growing oligomer before insertion happens. Notably, this calculation is based on the assumption that dimers do not join the growing oligomer, i.e. the probability would be considerably lower if joining of dimers and oligomers can occur (please see response to point #2 from reviewer #2 for further discussion).

Results (pg 13)

“Finally, we calculated the upper limit for the probability of nucleating a second dimer on a liposome before insertion of the growing PFO oligomer (Figure 6—figure supplement 5). This analysis showed that the majority of liposomes are predicted to contain a single oligomer at the time of dye release at concentrations ≤100 nM, but nucleation of a new dimer become probable at higher concentrations (note that this dimer may not form an independently growing oligomer but could join the first oligomer).”

5) Please address the recommendations raised by the reviewers below. These suggestions are directed at clarifying the manuscript and interpretation of the results.

Please see below.

Reviewer #1 (Recommendations for the authors):1. In Figure 1D, it will be useful to add examples of the "single step", "no step", and "other" traces directly in this main figure, as has been depicted in Figure 1 —figure supplement 3. Visualization of these representative traces is important to allow the reader to properly interpret the different behaviors.

As suggested, the figure has been revised to include examples of “no step” (Figure 1D) and “other” (Figure 1E) traces. The new panels are referenced in the Results section.

Results (pg 6)

“At AF647-PFO concentrations between 100–500 pM, most liposomes lost their dye signal in a single step (>50%) (Figure 1C and 1F), while ~30% of dye release traces showed partial or multi-step signal loss (Figure 1D and 1F). The remainder (~10%) showed no or little signal loss (Figure 1E and 1F), suggesting that these were not permeabilised despite AF647-PFO binding to many of these liposomes (further examples shown in Figure 1 Figure Supplement 3B).”

2. In 2G, the super-resolution approach for PFO monomer binding appears to outline the liposome size quite well. Can you quantify the liposome size distribution based on this approach? Do you find that there is a significant amount of non-uniformity to indicate that your results may depend on liposome size?

We have quantified the diameters of liposomes using super-resolution reconstruction. The resulting size distributions have been added to Figure 1 – Supplementary Figure 1 as panel B to allow for direct comparison to the dynamic light scattering (DLS) measurement. Overall, the super-resolution diameter measurement of 172 (±18) nm is consistent with the DLS measurement of 183 (±37) nm. We have revised the Results and Methods accordingly.

Results (pg 5)

“The liposomes had an average diameter of ~200 nm (183±37 nm measured by dynamic light scattering, 172±18 nm measured using super resolution microscopy; Figure 1—figure supplement 1) and were captured on the surface of a streptavidin-coated glass coverslip at the bottom of a microfluidic channel device.”

Methods (pg 22)

“Liposome diameter measurements using super-resolution TIRF microscopy. Super-resolution images were exported as a grayscale image from the Render Picasso program with a fixed 5-fold up-sampling (giving an effective pixel size of 22 nm). Particles were detected by thresholding and liposome sizes were calculated using a Feret diameter.”

3. Addition of uncertainty and statistical information. Much of the data presented lacks a description of the uncertainty or variability. For example, in the frequency/percentage bar plots, the binomial standard deviation could be used. In other plots, the mean is shown but without standard deviation. Please add this information and provide details in the legends about population or sample numbers.

We have updated the legends for all figures to specify the number of independent experiments for each condition.

We have updated plots that previously showed mean values (Figures 1F, 4E, 5E, 6D, 7A and 7G) to include individual data points derived from independent experiments to show the spread in the data.

Dot plots of the values derived from independent experiments for the kinetic parameters determined in this work are shown in Appendix Figure A2. Mean and standard deviation obtained from this analysis are listed in Table 1.

4. Revise the following paragraph on page 4 as this paper is no longer accompanying:"In an accompanying paper, Wallace and colleagues use single-molecule fluorescence tracking of PFO assembly on a droplet interface bilayer (Senior et al., 2021). These imaging studies support the insertion of incomplete arc-shaped membrane lesions, suggesting an alternative mechanism of pore formation that is distinct from the canonical prepore formation prior to insertion. This discrepancy raises the question as to when and how release of the membrane spanning regions is triggered, which cannot be correlated with key assembly steps using ensemble methods. Related to this matter, it is also unclear whether release and insertion of the membrane spanning regions from each of the subunits occurs in a concerted or sequential fashion."

We have revised this sentence.

Introduction (pg 4)

“Imaging modalities applied to CDC assembly on planar lipid bilayers include high speed atomic force microscopy, as shown for suilysin (Leung et al., 2014) and listeriolysin O (Ruan et al., 2016) and single-molecule fluorescence tracking, as shown for PFO assembly on a droplet interface bilayer (Senior et al., 2021). These imaging studies support the insertion of incomplete arc-shaped membrane lesions, suggesting an alternative mechanism of pore formation that is distinct from the canonical prepore formation prior to insertion. This discrepancy raises the question as to when and how release of the membrane spanning regions is triggered, which cannot be correlated with key assembly steps using ensemble methods. Related to this matter, it is also unclear whether release and insertion of the membrane spanning regions from each of the subunits occurs in a concerted or sequential fashion.

5. Further technical details would be useful in this paper, such as how the co-localization was determined and corrected for different mappings between fields, as well as how the number of PFO molecules was ascertained by intensity analysis.

Mapping between channels: Please see response to point #1 of the essential revisions.

Intensity analysis: Additional information on the quantification of single-molecule intensities has been added in the Appendix section “Single-Molecule photobleaching for intensity calibration”.

Reviewer #2 (Recommendations for the authors):1. Page 7. How was the surface area of the liposome established?

The surface area of a liposome was approximated using the surface area of a 200 nm sphere. The relevant sentence has now been reworded to state this.

Results (pg 7)

“This analysis allowed us to obtain the monomer binding rate constant (*B* = 0.9±0.23 nM^-1^ s^-1^) which corresponds to 7.2±1.8 nM^-1^ s^-1^ μm^-2^ when taking the surface area of liposomes into account (calculated as the surface area of a 200 nm sphere).”

2. Page 9-10. The authors go to a great length to describe the kinetics of oligomer formation. However, it is not clear how the authors can actually distinguish between the formation of many independent dimers and the formation of oligomers. Do they assume that once the dimers are on the membrane they will eventually meet up into oligomers? Is this a reasonable assumption given the diffusion time of the dimers and the size of the liposome?

Formation of many dimers vs oligomerisation?

1. The experimentally measured rate for the signal increase after nucleation is 6–20 times faster than dimerisation within the concentration range of our experiments (shown in the new Figure 6—figure supplement 5A and B). This suggests that the signal increase after nucleation is primarily driven by monomer addition to an existing oligomer rather than formation of new dimers.

2. The movement of the PFO species on liposomes shows a sharp transition from high to low upon poreopening (new Figure 7). This sudden drop in movement would not be observed if there were several independently diffusing species.

Can newly formed dimers join an existing oligomer?

As shown by tracking movements of PFO oligomers on liposomes (new Figure 7) and flat bilayers (Senior et al. 2021) dimers and oligomers in the prepore state diffuse readily on the membrane such that these entities can collide on the liposome surface. On the basis of these observations, we think that it is plausible that dimers could add to an existing oligomer on the membrane. Nevertheless, this remains an open question which requires further investigation.

Both of these questions are relevant in the context of whether a second PFO oligomer is likely to nucleate on the liposome when an actively growing oligomer is already present on the membrane. Please see our response to point #4 of the essential revisions for additional discussion.

3. Furthermore, it is not clear how the authors can distinguish between the formation of conductive arc pores and full pores. Since it is only possible to observe a full liposome, all these species should have the same fluorescence properties. How can the authors know the number of subunits of a pore, considering that the fluorescent signal can only estimate the overall number of units in the membrane? How can the authors distinguish a full pore from two arcs?

Please see response to point #2 of the essential revisions.

4. Page 10. The authors argue that the arc pores with at least four subunits can form pores. How did they come to this conclusion? Only on the basis of fluorescence intensity? If that is the case, has photobleaching taken into account? The authors identified the oligomer-length specific insertion rates from the insertion times. However, it is not clear how this was done. The authors should either point to the methods where more details should be given, or explain more in detail how this was done (e.g. is the number of subunit estimated by the fluorescence intensity?).

Has photobleaching been taken into account when using intensity for determining the minimum oligomer length required for insertion?

To avoid photobleaching, the experiments used to measure oligomer lengths were conducted as follows: (1) Assembly of AF647-PFO oligomers on liposomes in the absence of laser illumination. (2) Removal of AF647-PFO from solution (wash-out) to halt oligomer growth and start of imaging to measure the intensities of PFO oligomers and their insertion kinetics. This protocol ensures that the number of subunits is determined at a time when no photobleaching has occured. We note that a different protocol was used for Figure 5A. In this case, imaging commenced at the beginning of the experiment to acquire complete traces that also include the PFO build-up during the first stage. We have added a sentence to the Methods to clarify this point.

Methods (pg 23)

“Dual-colour TIRF imaging was conducted using a 488 nm laser and a 640 laser with 50 ms exposure time. To avoid photobleaching in experiments used to measure the number of subunits in oligomers, imaging only commenced following the washout of free PFO in solution”

How was the number of subunits determined from the fluorescence intensity?

Additional information on the quantification of single-molecule intensities has been added in the appendix (“Single-Molecule photobleaching for intensity calibration”).

5. The kinetic model (Figure 6B) also assumes that a monomer is added to a nucleated pore. However, since monomers are transient and dimers rather stable, would it make more sense that dimers are constantly formed from monomers and these structures move around the membrane until they meet each other and then form full pores? Would it also not be easier for a full pore to be formed from two arc pores? The authors should spend a few arguments to convince the reader that the experimental setting can distinguish between these different scenarios?

Formation of pores from dimers?

Please see response to point #2 from this reviewer.

Formation of pores by coalescence of two arc pores?

We cannot rule out the formation of coalesced arc pores on the basis of the fluorescence intensity data. This process would be dependent on the nucleation of a second independent PFO oligomer; please see response to point #4 of the essential revisions for further discussion.

However, we suggest that the formation of different types of structures (arc pores, coalesced arc pores and ring pores) is kinetically controlled (see Discussion). In the low concentration range used here (and due to the small surface area of liposomes), we think that nucleation of an independent second structure before insertion of the first structure is unlikely in our experiments such that coalesced arcs are presumably also unlikely. This is supported by the negative staining EM analysis of PFO structures on liposomes (Figure 6 Figure–Supplement 4) showing complete rings, but not coalesced arcs, consistent with our interpretation that arc pores continue to add monomers to form full rings.

6. Pore insertion kinetics (Page 12). This connects to the arguments above. How can the authors distinguish between the growing of arc pores from the formation of unrelated structures in the same liposome? Why this cannot be explained with a new nucleation process.

Please see our response to point #4 of the essential revisions.

7. The authors observed a marked pH dependence of pore formation. How is the fluorescent signal affected by the pH? Please also indicate where the reader can find the mathematical models that allow distinguishing between membrane and lateral interactions.

Dependence of AF647 intensity on pH:

We have measured the fluorescence intensity of the dye between pH 5–8 and found that it is constant in this range. This data has been added as Figure 8—figure supplement 2 and is referred to in the legend for Figure 8.

Legend for Figure 8.

“Control experiments confirmed that the intensity of AF647-PFO was independent of pH (Figure 8—figure supplement 2).”

Mathematical models:

An additional section has been added to the appendix (“The effect of pH on nucleation”) to provide details of these models.

8. Furthermore, the authors should measure directly (on and off rates) the monomer and dimer affinity for the liposomes at different pH values by fluorescence.

We agree that measuring on- and off-rates of the monomer as a function of pH would further validate our model. However, the focus of this paper is on a complete kinetic description of the PFO pore formation pathway, whereby the pH experiments were included in this paper as a case study to demonstrate the utility of the kinetic model. As such, further experiments dissecting specific perturbations of the pathway are beyond the scope of this study, especially since the monomer binding experiments are very time consuming.

9. Figures. Please add the condition (salt, buffer and pH) in the legend of the figures.

We have added buffer conditions to the legend of Figure 1. The buffer conditions remain the same throughout the remainder of the manuscript (apart from the changes in pH, which are specified in the legend of Figure 8).

“All experiments were conducted at room temperature using a 20 mM HEPES buffer (pH 7) containing 100 mM NaCl and 0.01 mg/mL BSA.”

Reviewer #3 (Recommendations for the authors):Overall, the quality of the experiments and their analysis is excellent, and I have only some questions to be addressed:1. The authors attribute the heterogeneous release profiles to heterogeneity in the liposome preparation. What about the stochastic nature of pore formation? That would also support the distributions observed between time of nucleation/poration.

Stochasticity/time distributions:

We absolutely agree that the pore formation process is inherently stochastic. Indeed, that is what necessitates studying the pore formation pathway at the single-molecule level, as each pore does not undergo the same process at the same time. As the reviewer points out, this dephasing leads to the distributions of times between different steps in the pathway.

Heterogeneous (multi-step) dye release profiles:

In addition to heterogeneity in the liposome preparation, multi-step dye release profiles could possibly arise if pore opening is transient, releasing only part of the dye before the membrane reseals. This would require the lifetime of the transient pore to be shorter than the time required for dye release. To get an understanding of the time scales involved in this process, we have simulated dye release via diffusion through pores of different diameters using FEM modelling. This analysis shows that dye release occurs on time scales of 100 μs to ~1 ms for pores with diameters between 35 nm and 2.5 nm. We have added the simulation of dye release as Figure 1—figure supplement 4 and added a sentence to the Results section (pg 6) referring to this new Figure.

Results (pg 6)

“For example, aggregated liposomes and multilamellar liposomes would be expected to give rise to traces with multiple steps or partial release of dye. Alternatively, multiple dye release steps could arise from transient pore opening events with a lifetime in the range of 0.1–1 ms (Figure 1—figure supplement 4).”

2. The fluorescence signal of PFO on individual liposomes seems heterogeneous, which the authors attribute to differences in cholesterol distribution. However, the lipid composition is not expected to present phase separation. Could this be due to complex assembly on part of the liposome leading to heterogeneous distribution? Otherwise, the authors might test their hypothesis of inhomogeneous cholesterol distribution by fluorescence microscopy on giant unilamellar vesicles or lipid bilayers containing a lipophilic dye.

We agree. We do not know what has caused these hot spots, and we should have been more circumspect in suggesting any underlying mechanism. The relevant section (pg 7) has been rewritten to reflect this:

Results (pg 7)

“Interestingly, many liposomes displayed bright spots in the reconstruction images suggesting that liposome membranes may have a spatially inhomogeneous affinity for binding PFO, but the underlying mechanism for this observation remains unclear.”

3. Please explain how the different kinetics of monomer addition to the pre-pore or the pore states are considered in the mathematical model.

The model contains steps only up to pore formation, i.e. monomer addition to the pore state is not part of the model.

4. Please explain how the oligomerization rate by monomer addition before and after insertion can be the same. One would expect that the energetics of the interactions of the monomers with the non-inserted and inserted oligomers would differ. This part of the model is not very convincing yet and requires elaboration.

We agree that addition of monomers to PFO oligomers post insertion requires further investigation to validate this finding and to identify the interactions that drive this process. One possible model to explain the observation that oligomerisation rates are the same before and after insertion is that oligomerisation is effectively a diffusion-controlled reaction. That is to say, the rate-limiting step in oligomerisation is a monomer diffusing close enough to the end of an arc to interact. If this is the case, then the difference in binding affinity of a monomer with an oligomer before and after insertion would have little impact on the observed oligomerisation rate. This scenario is plausible given the tight binding of monomers adding to a growing oligomer and the very low concentrations used in this study. We have not included this model in the paper as it is highly speculative.

5. Do the complete rings continue to grow indefinitely? Please explain.

Complete rings are stable structures and do not continue to grow. Our data nevertheless suggest that PFO can accumulate on liposomes after the number of subunits required to form a complete ring has been reached. This continued increase in PFO signal is likely due to the growth of an additional PFO oligomer on the same liposome; please see also our response to point #11 from this reviewer.

6. While the authors nicely confirm that AF647-labelled PFO is functional and that the effect of self-quenching is small using different ratios of labelled to unlabeled PFO (Figure 1 —figure supplement 2), this analysis also revealed that a 100% ratio of labelled PFO is more active in perforating liposomes. While this effect can be caused by an underestimation of the amount of unlabeled protein (as the authors mention), they should also discuss the option that AF647-PFO is indeed more active due to possible structural alterations caused by the label.

The relevant section in the figure legend now reads:

“The slight increase in the rate of poration in the 100% labelled sample (maroon curve) in comparison to lower labelled percentage samples suggests that the actual concentration of the labelled sample may be slightly higher than the unlabelled sample. Alternatively, this could be the result of fluorescent labelling increasing the activity of PFO. Regardless, the activity of all samples is reasonably similar and we conclude that the labelled PFO is functional.”

7. The authors use photobleaching step counting of sparsely distributed AF647-PFO immobilized on glass coverslips to measure the reference monomer fluorescence intensity (as shown in Figure 1E and D). Considering that AF647-PFO has an estimated labeling efficiency of ~1.5 fluorophores/PFO, it would be beneficial to compare the population distribution of the determined monomer intensities (Figure 1E) to the intensity distribution measured for monomeric AF647 dye immobilized on glass.

We analysed the first frame from photobleaching experiments with Picasso using the same parameters as in the liposome binding experiments, which should equally account for labelling efficiency. This is the distribution shown in Figure 2D.

8. In order to measure the oligomeric state of AF647-PFO at different time points in the liposomes, the authors determine the ratio of the fluorescence intensity associated with the liposome to the monomer fluorescence intensity derived from single photobleaching events. Given that not all PFO molecules are labelled with one single AF647 fluorophore (Figure 2E), the calculation of molecular units from fluorescence intensity is somewhat error prone. Labelling efficiency correction of the measured fluorescence intensities should be used to test if there is an influence of the labelling efficiency on the determined oligomeric state of AF647-PFO.

Please see response to point #2 of the essential revisions.

9. Considering the lipid composition of the outer leaflet of the plasma membrane (as a biological target for PFO pore formation), a concentration of 55% cholesterol seems very high. The authors should state in the manuscript why they chose the specific lipid mixture and how it might affect the mechanistic regulation of PFO (membrane binding, oligomerization, etc.) compared to the physiological scenario of the plasma membrane.

This mixture is frequently used in in vitro studies of PFO and we have used it here to facilitate comparison with the literature. We have added the following sentence to the Results section (pg 5):

Results (pg 5)

“The high cholesterol content facilitates PFO activity on model membranes and is consistent with previous biochemical studies (Shepard et al. 2000).”

10. When discussing the possible explanations why PFO pores continue to grow over time with wit a membrane insertion rate increasing with concentration, the authors should also discuss the effect of continuous protein insertion on the physical properties of a toroidal pore. Increasing number of protein molecules inserted in a membrane increase the membrane tension and at the same time decrease the line tension at the pore rim both promoting the stabilization of an open pore.

We agree that there are likely to be effects of membrane tension and line tension on the stability of an open pore. Since our method does not measure pore stability (i.e. closing of the pore is not detectable), we would prefer not to speculate on this aspect.

11. In this line, the authors should also include a figure showing the oligomerization characteristics of PFO (as in Figure 4 B) beyond the expected number of molecules required to form a full pore ring until the oligomerization reaches saturation. The possible explanation for the continuous increase in the number of PFO molecules that "eventually multiple pores form" (page 6) is contradictory to the valid assumption made later that an existing PFO oligomer in the liposome "acts as a sink for monomers" (page 12) that newly bind to the membrane.

Plots of the mean of all AF647-PFO intensity traces aligned at the time of nucleation (see Author response image 1) that extend beyond the range we would consider reliable (i.e. beyond growth of the first pore) show a slight increase in oligomerisation rates, consistent with multiple independently growing oligomers. We prefer not to include this analysis in the paper for the reasons stated in our response to point #4 of the essential revisions.

**Author response image 1. sa2fig1:** 

12. The authors should state in the manuscript why a Cysteine-free version of PFO (PFO(C459A)) was used.

We have added this information to the Methods section.

Methods (pg 19)

“PFO C459A lacks the cysteine residue prone to oxidation (leading to inactivation) and has the same pore-forming activity as wild type PFO (Pinkney et al. 1989; Saunders et al. 1989; Shepard et al. 1998).”

13. Quantify the percentage of unpermeabilized liposomes in Figure 2 —figure supplement 2.

We have quantified the proportion of unpermeabilized liposomes by counting the fraction of liposomes that retain at least 25% of their initial intensity. From this analysis, we have determined that on average 91% of liposomes remain unpermeabilized over the course of the experiment. This analyis has been added as panel D to Figure 2—figure supplement 2.

14. Please provide statistics (i.e. number of measured vesicles or complexes, number of technical replicates and test for statistical significance if applicable) for the data shown in Figure 1 D and E, Fig, 2 E and D, and Figure 1 —figure supplement 1.

We have added the requested information to the corresponding figure legends.

15. It is misleading that PFO oligomers with a number of subunits below the reported number needed to form a full ring are referred to as arcs. There is no experimental prove in this study that these oligomeric states of PFO indeed assemble to an arc-like structure in the membrane.

We agree and have added the following sentence (pg 10) to address this point:

Results (pg 10)

“While the fluorescence data do not provide structural information, we interpret these low stoichiometry PFO oligomers as arcs since wild type PFO assembly intermediates appear as arc-shaped structures in AFM images (Czajkowsky et al. 2004).”

16. The terminology "oligomer length" implies a structural parameter. Better refer to it as "molecular size" / "oligomeric state2 / "no of subunits/molecules".

As suggested, we have replaced the term “oligomer length” with “number of subunits”.

17. Please clarify that Figure 3D shows the sum of individual localizations over time and not multiple simultaneous localizations of AF647-PFO to one liposome.

The legend now reads:

“(C,D) Diffraction-limited TIRF image of the AF488 content dye (C) and corresponding map of all x/y-localisations over time of the single AF647-PFO species (D) that remains bound to the liposome for the duration of the peak in panel A. The scatter of localisations reflects the diffusion of the species on the liposome surface.”

18. The deliberate offset of the two fluorescence channels in Figure 2 —figure supplement 2 is somewhat misleading as it hinders the correct interpretation of the data. It would be better to show the correctly aligned image of the overlay and the single channel images and the merge of the cropped area for better visualization.

As suggested, Figure 2—figure supplement 1 now shows the correctly aligned images.

19. Please specify the meaning of the orange line in Figure 7 F and H.

Specified in the main text (pg 14) and the corresponding Figure legend (now Figure 8).

Results (pg 14)

“When we simply introduced the pH-dependent differences in the oligomerisation rate (Figure 8C) (which in turn result from the defect in membrane binding) into the oligomerisation/insertion kinetic model (Figure 6B), the model was able to reproduce the decrease in mean number of subunits (Figure 8F, orange line) and complete ring formation (Figure 8G, orange line) and the increase in the time from nucleation to poration (Figure 8H) with increasing pH (See appendix “The effect of pH on growth and insertion” for model details).”

Figure 8.

“The orange lines in panels F-H represent fits of the model in Figure 6B using the pH-dependent oligomerisation rates in panel C.”

20. Page 8: Please specify in the sentence "To determine the stoichiometry of AF647-PFO in the long-lived state, we measured the average intensity of all long-lived signals on liposomes." that it is averaging over time and not over several particles.

The sentence now reads:

Results (pg 8)

“To determine the stoichiometry of AF647-PFO in the long-lived state, we measured the average intensity over time of each long-lived liposome-bound signal.”

21. Please include scale bars in the microscopy images in Figure 1C, Figure 2C, Figure 2B, Figure 2 —figure supplement 1, and Figure 2 —figure supplement 2.

Scale bars have been added to Figure 2—figure supplement 1 and Figure 2—figure supplement 2. Given the small size of the snapshots in Figure 1C, 2C and 3B, we have stated the width of the snapshots in the corresponding legends to provide a reference for the scale of these images.

22. Refer to Table 1 also in the main text of the manuscript.

References to Table 1 added on page 7, 8, 9 and 10.

23. Page 13: Please correct the referencing to the different panels in Figure 7 F-I.

Corrected.

24. Page 6: Please correct the typos "Figur1 Figure Supplement 3" and "Figur1 Figure Supplement 3B".

Corrected.

25. Figure 3D: Please correct the typo "AF4647-PFO" in Figure 3D.

Corrected.